# When Data Is Scarce: Scaling Sparse Language Models with Repeated Training

Boqian Wu [* 1 2]  Qiao Xiao [* 3]  Patrik Okanovic [4]  Tomasz Sternal [4]  Maurice van Keulen [2]  Mykola Pechenizkiy [3]
Elena Mocanu [2]  Torsten Hoefler [4]  Decebal Constantin Mocanu [1]

## Abstract

Scaling laws for dense LLMs under infinite data are well explored, but how sparsity interacts with limited data is not. In this work, we study sparse training in data-constrained regimes where limited unique tokens require multi-epoch training. Our experiments span models up to 1.92B parameters in the fitting set, sparsity up to 93.75%, unique data budgets up to 2.6B tokens, and total training tokens up to 41.6B over 16 epochs; we further validate extrapolation on held-out dense-equivalent models up to 7.68B parameters. We find that: ① **Sparse scaling in data-limited settings:** We introduce a scaling law that models loss as a function of active parameters, unique tokens, data repetition, and sparsity, accurately predicting performance across compute and data budgets. ② **Delayed data saturation:** sparse training postpones diminishing returns from repeated data, making multi-epoch training more effective. ③ **Resource trade-offs:** With fixed data, loss-optimal sparsity is moderate ($\sim 50\%$), while compute-optimal sparsity is higher and grows with data scale. Overall, sparsity is not just a tool for efficiency, but a mechanism for improving scaling trade-offs under data scarcity. Our code is available at: https://github.com/boqian333/sparse-dc-scaling.

## 1. Introduction

Scaling has been central to the recent success of large language models (LLMs) (Kaplan et al., 2020; Brown et al., 2020; Hoffmann et al., 2022; Touvron et al., 2023a;b; Grattafiori et al., 2024; Jaech et al., 2024; Achiam et al., 2023; Bi et al., 2024; Anil et al., 2023; Yang et al., 2024). While claims about the "death" of AI scaling laws remain

actively debated [1], growing evidence suggests that scaling behavior continues to evolve rather than simply saturate (Cheng et al., 2026; Sadhukhan et al., 2026).

A key driver of this evolution is the growing mismatch between data availability and computation. While the supply of high-quality human data increases at a modest rate of approximately $1.03\times$ per year, the data used for pre-training has grown much faster, at roughly $4\times$ per year (Sevilla & Roldán, 2024; Villalobos et al., 2024). As a result, the exhaustion of high-quality data has ushered in a data-constrained pre-training regime (Villalobos et al., 2022; Muennighoff et al., 2023), fundamentally reshaping the landscape of LLM scaling. More broadly, this shift highlights a central research challenge: how to enhance model capabilities using a finite amount of high-quality data, rather than relying solely on increasing data volume.

Under data-constrained regimes, commonly adopted approaches for improving model performance, such as allocating additional compute to increase model size or repeating training data, often exhibit diminishing returns and incur substantial computational costs (Muennighoff et al., 2023; Kim et al., 2026). Although stronger regularization, model scaling, and ensembling can improve data efficiency in data-constrained pre-training (Kim et al., 2026), they largely operate within the conventional dense pre-training paradigm and do not fundamentally alter the underlying architecture or introduce a qualitatively different scaling mechanism.

This distinction motivates a broader question: under finite data, is scaling constrained only by the amount of model capacity, or also by how that capacity is distributed and optimized? In dense training, all parameters are updated at every step on the same limited data, which can accelerate saturation under repeated exposure. This raises the possibility that scaling efficiency may be improved by expanding the total parameter space while keeping per-token compute, determined by the number of non-zero parameters, fixed or even reduced. Such a perspective challenges the assumption of fully dense pre-training and motivates the use of sparsity

---

[1]University of Luxembourg [2]University of Twente [3]Eindhoven University of Technology [4]ETH Zürich. Correspondence to: Boqian Wu <boqian.wu@uni.lu>, Qiao Xiao <q.xiao@tue.nl>.

*Proceedings of the $43^{rd}$ International Conference on Machine Learning*, Seoul, South Korea. PMLR 306, 2026. Copyright 2026 by the author(s).

[1]See, e.g., public discussions on the potential limits and future evolution of scaling laws in widely read blogs (Sevilla et al., 2024; Marcus, 2025) and in coverage by major news outlets (Nolan, 2024; Varanasi, 2025).

(Mocanu et al., 2018; Frankle & Carbin, 2018; Evci et al., 2020; Hoefler et al., 2021; Wu et al., 2025).

Sparsity, however, can be introduced in different ways. Architectural sparsity, such as in Mixture-of-Experts (MoE) models (Shazeer et al., 2017), increases model capacity through conditional activation, but largely preserves dense optimization within each expert. In contrast, sparse training methods, such as Dynamic Sparse Training (DST) (Mocanu et al., 2018; Bellec et al., 2018; Mostafa & Wang, 2019; Evci et al., 2020; Yuan et al., 2021; Liu et al., 2021; Xiao et al., 2022; 2025; Wu et al., 2025), impose sparsity directly during optimization by updating only a subset of parameters and dynamically evolving the network structure. Architectural sparsity alone does not change how often the same parameters are optimized, whereas parameter-level sparse optimization redistributes updates across a larger pool of weights. Nevertheless, how sparse optimization reshapes the scaling behavior of LLM pre-training under data-constrained regimes remains insufficiently understood.

In this study, we seek to develop a systematic understanding of how DST scales in data-constrained LLM pre-training by formulating and analyzing its scaling behavior. To make this investigation tractable, we structure our investigation around three key questions:

**Sparse scaling law under data-constrained regimes.** How does pre-training performance scale with model size and data repetition across different sparsity levels under data constraints? This question is addressed in Section 3.

**Allocation.** How does sparsity influence the optimal trade-off between model size and data repetition under fixed data constraints and compute budgets? This trade-off is analyzed in Section 4.

**Optimal sparsity.** Given a fixed data budget, how can we determine the optimal sparsity level for efficient pre-training? The optimal sparsity strategy is studied in Section 5.

To answer these questions, we conduct experiments across sparsity, model size, data budget, and training duration, covering sparsity up to 93.75%, unique data budgets up to 2.6B tokens, and total training tokens up to 41.6B over 16 epochs. We fit scaling laws on models up to 1.92B dense-equivalent parameters and validate extrapolation on held-out models up to 7.68B parameters, leading to the following key findings:

① **Sparse training improves data utilization efficiency.** We find that sparsity increases the data saturation scale $R_d^*(S)$, which governs the decay rate and asymptotic headroom of effective data under scaling. $R_d^*(S)$ rises from 4.4 at $S = 0$ to about 6.90 at $S = 0.5$, implying slower saturation and a larger effective-data ceiling for sparse models.

② **Sparse training improves parameter efficiency.** Compared to dense training, sparse training benefits relatively more from allocating additional compute to longer training rather than to increasing parameter count. This leads to up to a $3\times$ smaller parameter-to-token ratio at a 1.3B-token data budget and compute on the order of $10^{20}$ FLOPs, improving inference efficiency.

③ **Compute-optimal sparsity enables cheaper large-scale training.** At compute-optimal sparsity, sparse models match dense-model accuracy while using approximately $8\times$ to $10\times$ fewer training FLOPs.

To our knowledge, we provide the first systematic study of DST under data-constrained scaling, showing that sparsity can improve scaling efficiency when data is limited.

## 2. Background and Motivation

Scaling laws (Kaplan et al., 2020; Hoffmann et al., 2022; Zhai et al., 2022; Muennighoff et al., 2023; Frantar et al., 2024) have played a central role in guiding the training of large language models by prescribing how compute should be allocated between model size and training data.

### 2.1. Chinchilla scaling law

A widely adopted model of dense scaling behavior is the Chinchilla scaling law (Hoffmann et al., 2022), which models validation loss as

$$L(N, D) = \frac{A}{N^\alpha} + \frac{B}{D^\beta} + E, \tag{1}$$

where $N$ denotes the number of model parameters and $D$ the number of training tokens. Here, $A$ and $B$ are positive fitted constants, $\alpha$ and $\beta$ are fitted scaling exponents, and $E$ denotes a fitted irreducible loss term. Under a standard Transformer cost model, the total training compute, measured in FLOPs, can be approximated as $C \approx 6ND$ (Kaplan et al., 2020; Hoffmann et al., 2022). Equation 1 predicts scaling trends in the infinite-data regime and yields a compute–optimal trade-off between parameters and tokens.

### 2.2. Two orthogonal perspectives on scaling

Two largely orthogonal lines of work characterize scaling behavior along different dimensions.

**Sparse scaling laws.** Sparse scaling laws extend dense formulations by explicitly accounting for the number of active parameters during training (Frantar et al., 2024; Jin et al., 2025). These works demonstrate that sparsity can improve parameter efficiency and enable larger effective model capacities under fixed compute budgets. However, they are typically studied in data-sufficient regimes and do not model data repetition or overfitting.

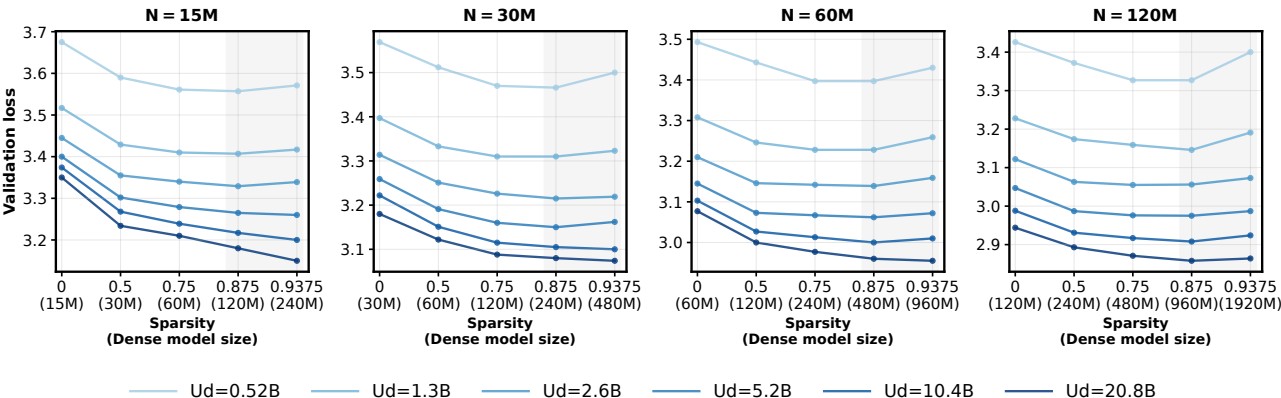

*Figure 1.* Sparsity–capacity trade-off in the data-constrained regime. Validation loss as a function of sparsity ($S$) for models with different numbers of non-zero parameters ($N = 15M, 30M, 60M, 120M$) and different unique-token budgets $U_d$.

**Data-constrained scaling laws.** Data-constrained scaling laws (Muennighoff et al., 2023) explicitly model diminishing returns from repeated training data by introducing the notion of effective dataset size $D'$.

$$D' = U_d + U_d R_d^* \left(1 - e^{-R_d/R_d^*}\right), \qquad (2)$$

where $U_d$ denotes the number of unique training tokens, while $R_d$ measures data repetition, with $R_d^*$ controlling the rate of diminishing returns from repeated tokens. This formulation models diminishing returns from repeated tokens as saturation in effective data, but assumes fully dense optimization and does not capture how alternative training paradigms such as sparse training may alter the value of repetition.

### 2.3. Why sparsity and data reuse interact non-trivially

The two lines of scaling laws described above have largely been developed in isolation; more details are provided in Appendix B. However, modern pre-training regimes often involve both sparsity (Agarwalla et al., 2024; Xiao et al., 2025; Ansell et al., 2024) and data reuse (Muennighoff et al., 2023; Kim et al., 2026), making it unclear whether insights from either line of work transfer directly to the other.

More importantly, sparse training alters not only model capacity but also how parameters are optimized. Dense training repeatedly updates the same parameters on limited data, accelerating saturation. Sparse optimization instead distributes updates across different subsets of parameters, potentially changing how models benefit from repeated data. Because scaling laws are highly non-linear, these factors interact in non-trivial ways, and dense data-constrained predictions may no longer apply.

To address this gap, we propose a framework that jointly studies sparse training and data reuse. To our knowledge, our work is the first to empirically characterize this regime

and clarify how sparse optimization reshapes scaling behavior under finite data.

## 3. Sparse Data-Constrained Scaling Law

Our goal is to derive a scaling law for dynamic sparse training (DST) under data-constrained regimes by characterizing how sparsity $S$ enters the data-constrained formulation. While prior work models sparsity and data repetition separately, it is unclear how sparsity simultaneously affects parameter efficiency, the value of repeated data, and over-fitting behavior. To guide the formulation, we first conduct exploratory experiments and distill empirical observations.

### 3.1. Experimental setup

We study the LLaMA-2 (Touvron et al., 2023b) family of models trained on the C4 dataset, and conduct extensive sweeps over sparsity, model size, data budget, and training epochs to derive scaling laws. We consider three distinct token budgets: 520M, 1.3B, and 2.6B tokens. For each budget, each model is trained for 1, 2, 4, 8, and 16 epochs, yielding total training token counts ranging from 520M to 41.6B in the repeated-token setting.

We vary the active model size across seven parameter targets, from 15M to 960M in $2\times$ increments: 15M, 30M, 60M, 120M, 240M, 480M, and 960M. Sparsity levels are selected from $\{0.0\%, 50\%, 75\%, 87.5\%, 93.75\%\}$, corresponding to compression factors from $1\times$ to $16\times$. To keep the sweep computationally tractable, we use a partial grid and omit the highest-sparsity settings for the largest active models. The resulting dense-equivalent model sizes, defined as $N/(1-S)$, where $N$ is the non-zero parameter count and $S$ denotes sparsity, range from 15M to 1.92B parameters. Our experimental suite consists of 500 training runs; we further validate extrapolation on held-out dense-equivalent models up to 7.68B parameters.

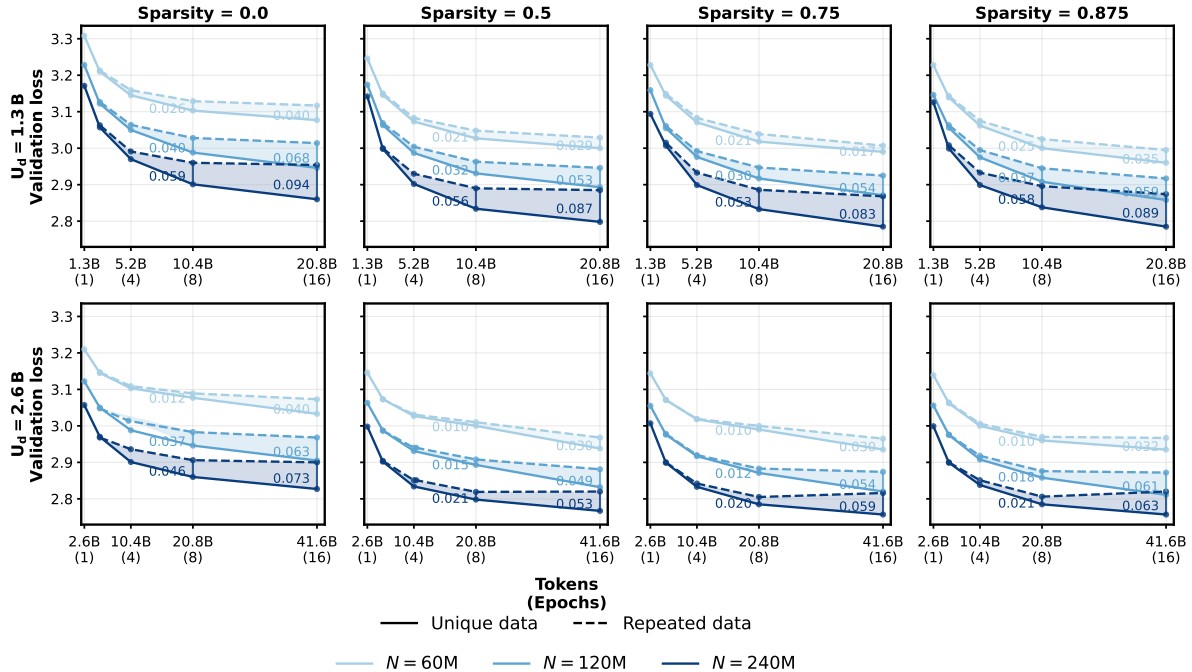

*Figure 2.* Scale-dependent effects of data repetition across sparsity levels. Validation loss as a function of training tokens (epochs) for models with different non-zero parameter budgets ($N = 60M, 120M, 240M$), training-token budgets, and sparsity levels. Solid lines correspond to training on unique tokens, while dashed lines correspond to training with repeated tokens.

All models are trained with a batch size of 512 using the Adam optimizer in bfloat16 precision. Learning rates are selected according to the $\mu$P scaling rule (Yang et al., 2021), with a linear warm-up applied over the first $1\%$ of training tokens. Sparsity is held constant throughout training, and masks are updated using magnitude pruning followed by random regrowth (Mocanu et al., 2018). Details of the DST algorithm are provided in Appendix C. This setup prioritizes consistency across scales rather than optimizing performance for any single configuration.

### 3.2. Sparsity under Data Constraints without Repetition

Before integrating data repetition, we isolate the effect of sparsity on the Chinchilla scaling law. While prior studies (Frantar et al., 2024; Jin et al., 2025) have explored sparse scaling laws, we re-examine this relationship beyond the standard compute-optimal infinite-data paradigm to account for the data-constrained regime. Figure 1 visualizes validation loss as a function of sparsity $S$ across different non-zero parameter and compute budgets. Across all settings, validation loss decreases mildly with increasing sparsity and then either plateaus or slightly increases at higher sparsity levels. In Figure 1, $N$ denotes the number of non-zero active parameters. The corresponding dense-equivalent model size is given by $N/(1 - S)$, which increases rapidly with sparsity. As a result, even for small values of $N$, high sparsity levels correspond to dense models with hundreds of millions to

billions of parameters. Therefore, the experiments in Figure 1 operate beyond the small-model regime and should not be interpreted as small-scale experiments. Our formulation is guided by the following observation as shown in Figure 1:

> **Observation 1: Sparsity–capacity trade-off**
>
> For a fixed budget of non-zero parameters ($N$), increasing sparsity ($S$) initially improves validation loss. However, these gains eventually saturate or diminish as $S$ reaches extreme levels.

> **Modeling implication**
>
> To capture this trade-off, we modulate the parameter-scaling term by a sparsity-dependent factor:
>
> $$\frac{A}{N^\alpha} \longrightarrow \frac{A\big((1 - S)^\epsilon + PS^\mu\big)}{N^\alpha}. \qquad (3)$$
>
> This induces a U-shaped dependence on $S$ and recovers dense scaling when $S = 0$.

Importantly, these results do not contradict prior sparse scaling laws (Frantar et al., 2024), but rather extend them to the data-constrained regime. Existing formulations (Frantar et al., 2024) are derived under a compute-optimal assumption with effectively infinite data, under which performance

improves monotonically with sparsity for a fixed number of non-zero parameters $N$. In the infinite-data regime, higher sparsity corresponds to a larger dense-equivalent model whose additional capacity can be fully exploited.

In contrast, under data-constrained training with unique data, the available data does not scale with the dense-equivalent model size. While increasing sparsity continues to increase effective capacity and can still reduce validation loss, the marginal benefit of additional sparsity diminishes as $S$ becomes extreme. At high sparsity levels, the extra capacity introduced by sparsity cannot be fully utilized given the fixed data budget, leading to saturation or mild degradation in performance.

This behavior manifests as the observed U-shaped dependence of validation loss on sparsity and reflects a capacity–data mismatch at extreme sparsity, rather than a breakdown of scaling with $N$. Consistent with this interpretation, when $N$ is sufficiently small, the available data remains adequate relative to the effective model capacity even at high sparsity levels, and validation loss continues to decrease monotonically with sparsity, as shown in Figure 1.

The analysis above isolates the effect of sparsity when each token is seen only once. In this regime, sparsity primarily acts as a mechanism for reallocating effective model capacity under a fixed non-zero parameter budget. However, capacity reallocation alone does not fully characterize practical data-constrained training. In many realistic scenarios, additional unique data is unavailable, and models are trained for multiple epochs over the same dataset. To address this question, we compare how dense and sparse training differ in their ability to utilize repeated data.

Figure 2 illustrates how models trained on repeated data compare to those trained on an equivalent amount of unique data across different scales. Across sparsity levels, we observe a consistent performance gap: models trained on repeated data plateau earlier, exhibiting diminishing returns with additional compute. For a fixed training data budget and repetition ratio, this gap increases with model size $N$, indicating that larger models experience diminishing returns from repetition more rapidly. Conversely, for a fixed model size, increasing the unique-token budget $U_d$ reduces the gap, as additional training allows repeated data to remain useful for longer.

These trends closely align with previous work on data-constrained scaling in dense networks (Muennighoff et al., 2023; Ni et al., 2025); a detailed discussion is provided in Appendix D.3. Motivated by this connection, we adopt the dense data-constrained scaling formulation of Muennighoff et al. (Muennighoff et al., 2023) as our starting point. In this formulation, repeated training does not contribute linearly to model performance. Instead, the loss is expressed in terms of an effective data size $D'$ and an effective number of parameters $N'$:

$$L(N', D') = \frac{A}{(N')^\alpha} + \frac{B}{(D')^\beta} + E, \qquad (4)$$

where the effective data size $D'$ captures the diminishing returns from repeated tokens. It is defined as

$$D' = U_d + U_d R_d^* \left(1 - e^{-R_d/R_d^*}\right), \qquad (5)$$

where $U_d$ denotes the number of unique training tokens, $R_d$ denotes the data repetition ratio, and $R_d^*$ controls how quickly repeated data saturates. A larger $R_d^*$ indicates slower saturation, meaning that repeated data remains useful for longer. Similarly, the effective number of parameters $N'$ captures diminishing returns from increasing model capacity and is defined as

$$N' = U_n + U_n R_n^* \left(1 - e^{-R_n/R_n^*}\right), \qquad (6)$$

where $U_n$ denotes the unique parameter budget, $R_n$ denotes the parameter scaling ratio, and $R_n^*$ controls the saturation rate of additional model capacity.

Equations 4 – 6 apply only to dense models. We next examine how introducing sparsity modifies the notion of effective data, and how this, in turn, influences the impact of data repetition. Figure 2 shows that sparsity reduces the performance gap between training on repeated and unique data, with the magnitude of this effect depending on the sparsity level. In particular, the gap narrows most noticeably at intermediate sparsity levels (around 0.5), where repeated data saturates more slowly than in dense models. At higher sparsity levels, the gap remains smaller than in dense networks, but the additional reduction is more modest.

> **Observation 2: Sparsity delays data saturation**
>
> Sparsity delays repeated-data saturation, with the largest gains at moderate sparsity and diminishing returns thereafter.

This observation suggests that sparsity shifts the data saturation point. We therefore explicitly model the data saturation threshold $R_d^*$ as sparsity-dependent.

> **Modeling implication**
>
> To capture this interaction, we allow the data saturation threshold $R_d^*$ to depend on sparsity,
>
> $$R_d^*(S) = R_d^* \left(1 + \lambda_1 S + \sigma_1 S^2\right), \qquad (7)$$
>
> which recovers the dense baseline at $S = 0$.

Within the explored sparsity range, this curvature represents a secondary correction to the leading linear effect, reflecting

diminishing marginal benefits of increased sparsity rather than a strongly non-monotonic dependence.

### 3.3. Sparse Data-Constrained Scaling Law

As in previous work (Muennighoff et al., 2023), we use a symmetric formulation for $R_n^*$, which is given by

$$R_n^*(S) = R_n^* \left(1 + \lambda_2 S + \sigma_2 S^2\right). \tag{8}$$

---

**Modeling implication**

The final sparse data-constrained scaling law:

$$L(N', D', S) = \frac{A\left((1-S)^\epsilon + PS^\mu\right)}{N'^\alpha} + \frac{B}{D'^\beta} + E. \tag{9}$$

The effective model size is defined as

$$N' = U_n + U_n R_n^*(S) \left(1 - e^{-R_n/R_n^*(S)}\right), \tag{10}$$

$$D' = U_d + U_d R_d^*(S) \left(1 - e^{-R_d/R_d^*(S)}\right), \tag{11}$$

with sparsity-dependent $R_d^*(S)$ and $R_n^*(S)$:

$$R_d^*(S) = R_d^* \left(1 + \lambda_1 S + \sigma_1 S^2\right), \tag{12}$$

$$R_n^*(S) = R_n^* \left(1 + \lambda_2 S + \sigma_2 S^2\right). \tag{13}$$

---

All coefficients are learned via L-BFGS following (Hoffmann et al., 2022), as described in Appendix D.2. The final fitted result is shown in Figure 3. Although the model includes multiple fitted coefficients, it generalizes well to unseen configurations (validation $R^2 = 0.914$ vs. fitted $R^2 = 0.982$), suggesting that the formulation captures real structure rather than overfitting.

The fitted coefficients provide insight into how sparsity affects both parameter efficiency and data reuse. In the parameter-scaling factor $(1-S)^\epsilon + PS^\mu$, the fitted value $P < 0$ shows that sparsity reduces the effective parameter-scaling penalty, improving parameter efficiency under a fixed non-zero parameter budget. Since $\mu = 0.847 < 1$, this benefit grows sublinearly with sparsity and therefore exhibits diminishing marginal returns. Meanwhile, $\epsilon \approx 0$ but slightly negative, so $(1-S)^\epsilon$ introduces a weak penalty near extreme sparsity rather than allowing sparsity to provide unbounded gains.

The fitted saturation thresholds reveal a complementary effect. Since $\sigma_1 < 0$ and $\sigma_2 < 0$, both $R_d^*(S)$ and $R_n^*(S)$ are concave functions of sparsity. However, $R_d^*(S)$ peaks later, around $S \approx 0.66$, increasing from $4.4$ at $S = 0$ to about $6.9$ at $S = 0.5$, which indicates that moderate sparsity delays repeated-data saturation. In contrast, $R_n^*(S)$ peaks earlier, around $S \approx 0.34$, and then decreases sharply, show-

ing that excessive sparsity limits effective model scaling. Together, these effects explain why moderate sparsity yields the largest gain: it improves parameter efficiency and data reuse while avoiding the model-capacity limitations that appear at high sparsity.

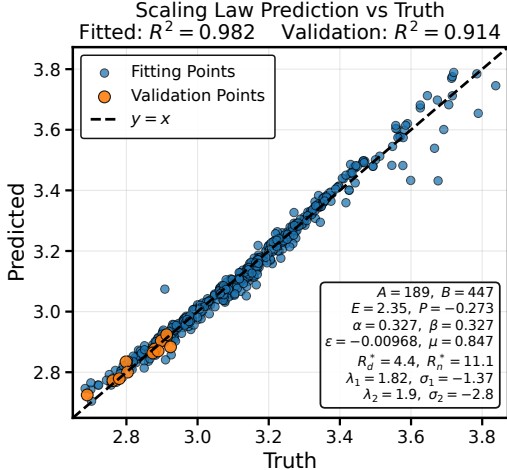

*Figure 3.* Prediction versus ground-truth loss. Each point corresponds to a trained model configuration.

## 4. Resource Allocation

### 4.1. Iso-Loss Curve

In our first experimental setting, we study scaling behavior under a fixed data budget for both sparse and dense training. With the amount of training data held constant, we vary compute allocation either by increasing the number of model parameters or by training for more epochs on the same dataset. Figures 4 (a) and (b) show the main results for scaling with 1.3B unique tokens for the dense case ($S = 0$) and the sparse case ($S = 50\%$). In the dense training setting (Figure 4 (a)), the data-constrained efficient frontier, corresponding to multi-epoch training, differs from the Chinchilla-style compute-optimal frontier derived under a single-epoch (no data reuse) assumption. Specifically, the data-constrained frontier shifts toward larger model sizes and fewer training epochs, indicating that once data is reused, allocating additional compute to model parameters becomes more beneficial than allocating it to further passes over the same dataset. This geometric shift of the frontier reflects the faster decay in the marginal value of repeated data relative to that of additional parameters (i.e., $R_d^* < R_n^*$).

In contrast to dense training, at 50% sparsity (Figure 4 (b)) the efficient frontier shifts to favor allocating additional compute to more training epochs rather than to increasing model parameters. For example, this behavior can be observed at compute levels on the order of $10^{20}$ FLOPs. At this scale, the near-optimal configuration in the dense case (Figure 4 (a)) is around 8 epochs with roughly 2B non-zero

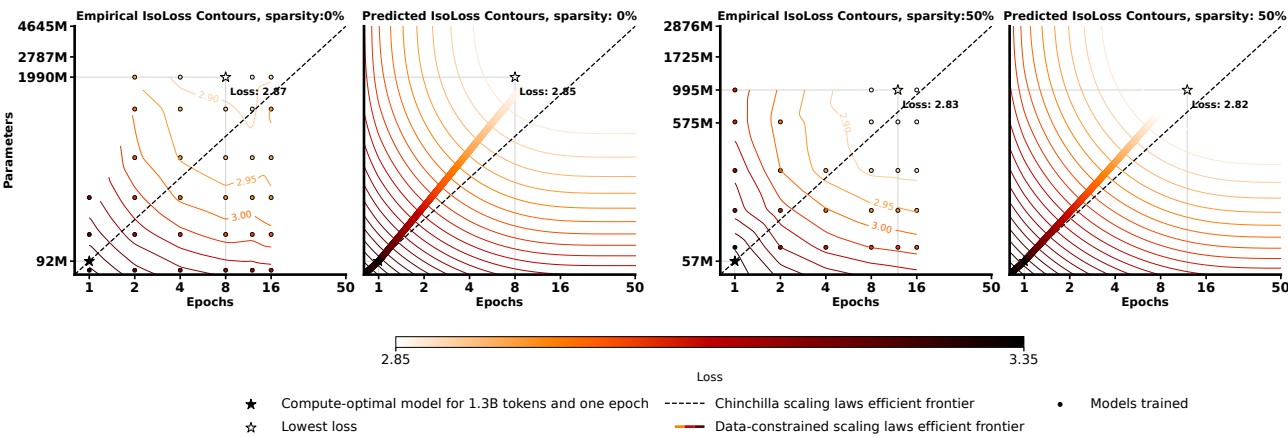

*Figure 4.* Iso-Loss curves under a fixed budget of 1.3B unique tokens. Top row: dense training ($S = 0$). Bottom row: sparse training ($S = 50\%$). Left column: empirical contours. Right column: predicted contours from the fitted scaling law.

parameters, whereas in the sparse case (Figure 4 (b)) the efficient frontier instead favors training for about 12 epochs with roughly 1B non-zero parameters. As a result, the frontier moves closer to the optimal allocation observed in the single-epoch training regime. This shift provides an additional advantage: smaller parameter-to-token ratios. For dense training, $N/D = 2\text{B}/(1.3\text{B} \times 8) \approx 0.19$, whereas for sparse training, $N/D = 1\text{B}/(1.3\text{B} \times 12) \approx 0.064$. This corresponds to an approximately $3\times$ reduction in the parameter-to-token ratio and, more importantly, uses fewer non-zero parameters, improving inference efficiency. This behavior reflects the sparsity-dependent saturation thresholds: moderate sparsity increases $R_d^*(S)$, allowing models to tolerate more data reuse, while excessive sparsity reduces $R_n^*(S)$ and limits effective model scaling. As a result, moderate sparsity shifts the optimal allocation toward longer training rather than larger non-zero parameter counts, consistent with prior sparse scaling laws (Frantar et al., 2024). In the data-constrained setting, this appears as delayed diminishing returns from data reuse.

### 4.2. Iso-FLOPs Curve

Before analyzing Iso-FLOPs behavior, we clarify why we consider two notions of training compute. For sparse models, training cost can be quantified either in terms of *sparse FLOPs*, which count only operations associated with non-zero parameters, $C = 6ND$, or *dense FLOPs*, which correspond to the cost of training a dense-equivalent model of size $N/(1-S)$. The former adopts an idealized algorithmic view, where per-token computation scales with the number of activated parameters rather than total dense model size, whereas the latter reflects the practical reality that current sparse training implementations often fall short of achieving ideal hardware-level speedups. By evaluating both definitions, we disentangle the effects of sparsity as a modeling choice from system-level considerations, and assess whether

our conclusions depend on how compute is measured.

As shown in Figure 5, across both sparse and dense FLOPs definitions, we observe a consistent compute-optimal regime at intermediate sparsity levels. While high sparsity yields faster gains at low compute under sparse FLOPs, its performance saturates more rapidly as compute increases due to data-reuse limitations. Conversely, dense models benefit from faster early optimization under dense FLOPs, but suffer from earlier saturation under repeated data. In both cases, moderate sparsity achieves the best balance between effective capacity and data reuse, leading to the lowest achievable loss at higher compute budgets.

> **Takeaway**
>
> Dense models favor more parameters, while sparse models benefit more from longer training with more epochs, and achieve better overall performance.

## 5. Optimal sparsity

Analyzing optimal sparsity elevates sparsity from a discrete design choice, as commonly treated in Section 4, to a quantity that can be systematically selected under fixed compute and data budgets. We characterize sparsity optimality under compute constraints through two complementary notions of optimal sparsity, corresponding to different optimization objectives.

**Loss-optimal sparsity.** The loss-optimal sparsity (highlighted by the yellow star in Figure 6) is defined as the sparsity level that minimizes validation loss under a fixed unique data budget. Formally, it is obtained by jointly optimizing model size, training duration (data repetition), and sparsity, without imposing any explicit constraint on training

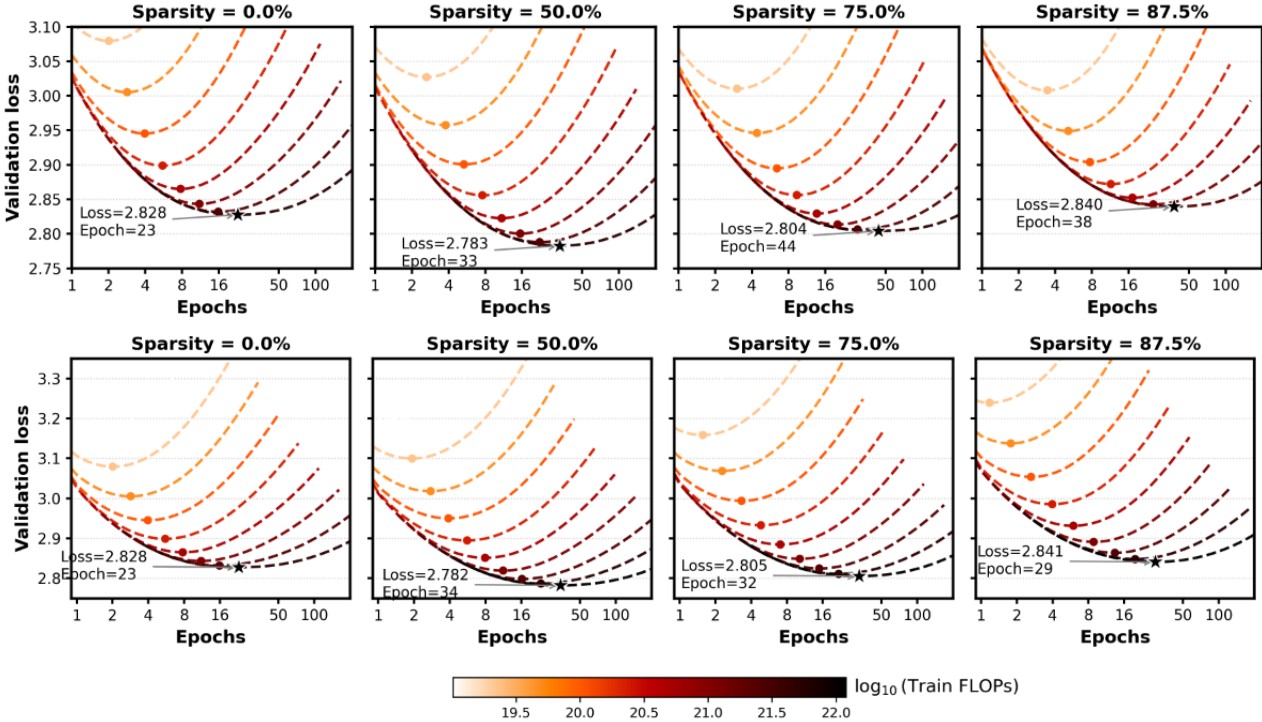

*Figure 5.* Iso-FLOPs curves across sparsity levels with a fixed budget of 1.3B unique training tokens. Top row: FLOPs are computed under a sparse cost model, $C = 6ND$, where $N$ denotes the number of non-zero parameters. Bottom row: FLOPs are computed under a dense cost model, corresponding to training a dense base model of size $N/(1-S)$ for the same number of training steps. Each curve traces validation loss as a function of training epochs at fixed compute, with colors indicating total training FLOPs.

compute:

$$S^*_{\text{loss}}(U_d) = \arg \min_{S \in [0,1)} \left( \min_{\substack{N>0 \\ D>0}} \mathcal{L}(N,\ U_d,\ D,\ S) \right). \quad (14)$$

**Compute-optimal sparsity at dense-equivalent performance.** The second notion is the compute-optimal sparsity, defined as the sparsity level that matches the globally optimal dense-model loss while minimizing training compute. Concretely, we first identify the minimum loss achievable by dense training, and then select the largest sparsity at which this loss is reached, while requiring less computation. This operating point represents (highlighted by the red star in Figure 6) the most compute-efficient configuration that preserves dense-level performance.

Let

$$\mathcal{L}^*_{\text{dense}}(U_d) = \min_{\substack{N>0 \\ D>0}} \mathcal{L}(N, \min(U_d, D), D, 0) \quad (15)$$

denote the globally optimal loss achievable by dense training. The compute-optimal sparsity is then given by

$$S^*_{\text{comp}}(U_d) = \arg \min_{S \in [0,1)} C \text{ s.t. } \mathcal{L}^*(C, S; U_d) \leq \mathcal{L}^*_{\text{dense}}(U_d). \quad (16)$$

We analyze how two different notions of optimal sparsity vary with the unique token budget ($U_d$). As $U_d$ increases from 1.3B to 13B and 130B tokens, the compute-optimal sparsity exhibits a clear upward trend, shifting from approximately $S \approx 0.6$ at $U_d = 1.3$B, to $S \approx 0.7$ at $U_d = 13$B and $U_d = 130$B. In contrast, the loss-optimal sparsity remains relatively stable across token budgets, consistently occurring in a moderate range around $S \approx 0.45-0.5$. Overall, while the sparsity level that minimizes loss varies little with $U_d$, larger token budgets increasingly favor higher sparsity when optimizing for compute efficiency at dense-equivalent performance.

Another consequence of this trend is improved parameter efficiency at scale. At the compute-optimal sparsity (red star), models achieve dense-equivalent performance with substantially fewer non-zero parameters than the dense-optimal baseline (Figure 6). For example, at $U_d = 1.3$B, the compute-optimal model uses $N \approx 0.68$B parameters versus $N \approx 2.60$B for the dense optimum, while reducing training compute by about $8\times$. This gap widens at larger scales: at $U_d = 13$B and $U_d = 130$B, compute-optimal models use $N \approx 6.92$B and $N \approx 70.71$B parameters, compared to $N \approx 27.06B$ and $N \approx 319.71$B for the corresponding dense optima, respectively. Overall, dynamic sparse training

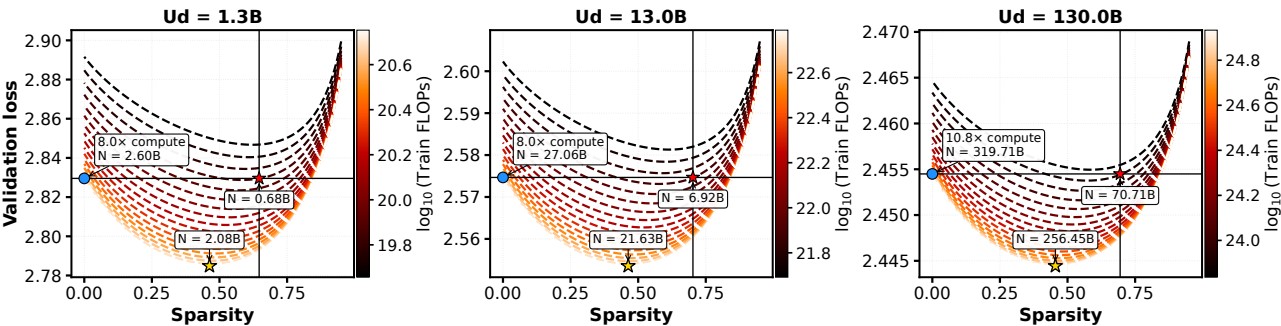

*Figure 6.* Optimal sparsity under the fixed unique-data budget with 1.3B, 13B and 130B unique tokens. With compute-optimal sparsity, the sparse model achieves dense-equivalent performance with fewer non-zero parameters than the dense optimum (e.g., 0.68B vs. 2.6B at $U_d = 1.3$B, and 6.92B vs. 27.06B at $U_d = 13$B).

yields models that are both compute- and parameter-efficient as data scale increases.

> **Takeaway**
>
> Loss-optimal sparsity stays moderate ($S \approx 0.45$–$0.5$), while compute-optimal sparsity rises with data scale to about $S \approx 0.6$–$0.75$.

## 6. Conclusion

We investigated the scaling behavior of Dynamic Sparse Training (DST) in data-constrained pre-training regimes, where limited unique data necessitates repeated training. Through extensive experiments spanning model size, sparsity, data budgets, and training duration, we showed that sparsity systematically reshapes the interaction between model capacity and data repetition. We proposed a sparsity-aware scaling law that jointly models active parameters, data reuse, and sparsity, accurately capturing performance trends beyond existing dense or sparse formulations. Empirically, we find that sparsity slows the saturation of repeated data, thereby enabling more effective multi-epoch training under fixed data budgets. Our analysis for the sparse data-constrained scaling law further shows that sparsity shifts optimal compute allocation toward longer training through data reuse rather than simply increasing model size, resulting in a smaller parameter-to-token ratio and consequently improved inference efficiency. Overall, these findings provide guidance for training large language models when data is the primary bottleneck.

## Impact Statement

This work analyzes how dynamic sparse training (DST) affects scaling under data scarcity and proposes a sparsity-aware scaling law for data-constrained pre-training. The results suggest that moderate sparsity can improve training efficiency and make better use of limited datasets, poten-

tially reducing compute and energy costs and benefiting domains with scarce data. We do not foresee any negative societal impacts arising from this research. Instead, we believe our findings will contribute to the broader adoption of efficient and sustainable AI models, ensuring their effectiveness in diverse and demanding environments.

## Acknowledgements

This work was partly supported by H2020 SmartCHANGE, grant agreement No. 101080965 and TTW Perspectief MegaMind projects. This work was carried out partly using the Dutch national e-infrastructure, with the support of the SURF Cooperative under grant nos. EINF-14276 and EINF-13990, and partly using the Luxembourg national supercomputer MeluXina. The authors gratefully acknowledge SURF and LuxProvide for their expert support.

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

| Symbol | Description |
|---|---|
| $C$ | Training FLOPs |
| $N$ | Total number of non-zero model parameters |
| $N'$ | Effective number of non-zero parameters |
| $D$ | Total number of training tokens processed |
| $D'$ | Effective number of training tokens |
| $U_d$ | Number of unique training tokens |
| $R_d$ | Number of data repetitions beyond the first epoch |
| $U_n$ | The optimal number of non-zero parameters for $U_d$ |
| $R_n$ | Parameter repetition factor |
| $A, B, E, \alpha, \beta$ | Fitted parameters for dense scaling |
| $\epsilon, P, \mu$ | Fitted parameters for DST without repetition |
| $R_d^*$ | Fitted saturation threshold for repeated data |
| $R_n^*$ | Fitted saturation threshold for excess model sizes |
| $\lambda_1, \sigma_1$ | Fitted parameters for data saturation |
| $\lambda_2, \sigma_2$ | Fitted parameters for model saturation |

*Table 1.* Summary of notation used throughout the paper.

## A. Notations

## B. Positioning Within Prior Scaling Regimes

Prior scaling law studies can be roughly categorized along two dimensions: model sparsity and data regime (data-sufficient vs. data-constrained).

- **Dense training + abundant data:** Well studied in classical neural scaling laws (Kaplan et al., 2020; Hoffmann et al., 2022; Zhai et al., 2022; Muennighoff et al., 2023; Frantar et al., 2024).

- **Dense training + limited data:** Studied in recent data-constrained scaling law work (Muennighoff et al., 2023; Ni et al., 2025).

- **Dense to sparse training + abundant data:** Explored in prior sparse scaling law literature (Frantar et al., 2024; Jin et al., 2025).

- **Dense to sparse training + limited data:** Largely unexplored.

- **Sparse to sparse training + abundant data:** Largely unexplored.

- **Sparse to sparse training + limited data:** Largely unexplored.

This paper focuses on the final regime: **sparse to sparse training under data constraints**, where dynamic sparsity interacts with repeated exposure to limited data. We provide the first systematic empirical characterization of scaling behavior in this setting.

## C. Preliminary on Dynamic Sparse Training

Dynamic Sparse Training (DST) methods aim to train sparse neural networks from scratch by dynamically evolving their sparse connectivity during training. Given a dense neural network $f_{\boldsymbol{\theta}}$ with parameters $\boldsymbol{\theta} \in \mathbb{R}^N$, a sparse version is obtained by applying a binary mask $\boldsymbol{m} \in \{0, 1\}^N$, yielding the sparse model $f_{\boldsymbol{\theta} \odot \boldsymbol{m}} \triangleq f_{\boldsymbol{\theta}'}$, where $\odot$ denotes element-wise multiplication. The sparsity level of the resulting network is defined as

$$S = 1 - \frac{\|\boldsymbol{\theta}'\|_0}{\|\boldsymbol{\theta}\|_0},$$

where $\|\cdot\|_0$ denotes the $\ell_0$ norm, which counts the number of non-zero elements.

Given a dataset $\mathcal{D} = \{(x_i, y_i)\}_{i=1}^N$, DST optimizes the sparse network $f_{\boldsymbol{\theta}'}$ to minimize the empirical loss:

$$\sum_{i=1}^N \mathcal{L}(f_{\boldsymbol{\theta}'}(x_i), y_i).$$

Let $\theta^l \in \mathbb{R}^{N^l}$ denote the parameters of the $l^{\text{th}}$ layer, where $N^l$ is the number of trainable weights in that layer **before masking**. The associated binary mask is $m^l \in \{0, 1\}^{N^l}$. Each layer has its own sparsity level $s^l \in (0, 1)$, defined as:

$$s^l = 1 - \frac{\|\theta^l \odot m^l\|_0}{N^l},$$

i.e., the fraction of weights in layer $l$ that are set to zero by the mask. This formulation ensures that both per-layer and global sparsity are precisely controlled, enabling consistent computational efficiency and training dynamics.

### C.1. Layer-wise Sparsity Allocation

Before training begins, a fraction of the model's parameters is masked to zero to match a predefined sparsity. A simple initialization scheme is the **uniform strategy**, which randomly generates sparse masks such that each layer has the same sparsity. All layers (except the first) are assigned equal sparsity $s^l = S$. The first and last layer are typically kept dense due to its high sensitivity.

### C.2. Prune-and-Regrow

DST periodically updates the binary mask $m$ through a **prune-and-regrow** process. This mechanism dynamically removes and introduces connections, allowing the network to adapt its sparse structure during training while maintaining a fixed sparsity level. In DST, the evolution of the sparse connectivity is governed by a predefined update schedule, which determines when and how the binary mask $m$ is modified during training. The schedule is controlled by the following hyperparameters:

- $\Delta T$: the number of training iterations between consecutive mask updates,

- $T$: the iteration after which mask updates are stopped,

- $\beta$: the initial fraction of connections to be updated at each step,

- $f_{\text{decay}}$: a decay function that modulates the fraction of updated connections over time.

The decay function $f_{\text{decay}}$ is applied every $T$ iterations until $T$ to gradually reduce the number of modified connections as training progresses. The function is defined as:

$$f_{\text{decay}}(t; \beta, T) = \frac{\beta}{2} \left( 1 + \cos\left(\frac{t\pi}{T}\right) \right),$$

where $t$ denotes the current training iteration. This schedule allows for aggressive exploration of sparse topologies early in training, while gradually stabilizing the connectivity pattern as convergence approaches.

There are two representative DST algorithms:

**Sparse Evolutionary Training (SET)** (Mocanu et al., 2018): At each update step, a fraction $p$ of the active weights with the smallest magnitudes is pruned, and an equal number of new weights are randomly reactivated and initialized to zero.

**Rigged Lottery Tickets (RigL)** (Evci et al., 2020): Also prunes the smallest-magnitude weights, but regrows new connections based on the gradients of inactive weights—specifically selecting those with the largest absolute gradient values.

The two algorithms show similar performance, but one incurs nearly double the computational cost in the scaling experiments. **Therefore, we adopt SET for this study.**

### C.3. Difference with Pruning, Dense-to-Sparse Training

DST begins with a randomly initialized sparse network and maintains **sparse throughout training**, dynamically evolving the connectivity via mechanisms such as prune-and-regrow. Both the weight values and the sparse topology are optimized simultaneously during training. This contrasts sharply with post-training and dense-to-sparse approaches, which inherit their initial topology from dense networks.

*Table 2.* Learning-rate sweep for dense training: LLaMA-120M, 240M, 480M, and 960M are trained with 2.6B tokens, while LLaMA-1.92B is trained with 5.2B tokens.

| Model size | Learning rate | Perplexity |
|---|---|---|
| Llama-120M | $2^{-9.5}$ | 23.12 |
| Llama-120M | $2^{-9.75}$ | **22.70** |
| Llama-120M | $2^{-10}$ | 22.81 |
| Llama-240M | $2^{-9.75}$ | 21.64 |
| Llama-240M | $2^{-10}$ | **21.27** |
| Llama-240M | $2^{-10.25}$ | **21.27** |
| Llama-480M | $2^{-10.25}$ | 20.45 |
| Llama-480M | $2^{-10.5}$ | **20.13** |
| Llama-480M | $2^{-10.75}$ | 20.32 |
| Llama-960M | $2^{-10.5}$ | 19.98 |
| Llama-960M | $2^{-10.75}$ | **19.71** |
| Llama-960M | $2^{-11}$ | 19.95 |
| Llama-1.92B | $2^{-10.75}$ | 17.48 |
| Llama-1.92B | $2^{-11}$ | **17.40** |
| Llama-1.92B | $2^{-11.25}$ | 17.79 |

### C.4. Learning Rate Selection

We determine the learning rate in two steps. First, we identify the learning-rate scaling rule for dense training across different model sizes. Then, we study whether this rule can be directly transferred to dynamic sparse training.

**Dense training learning rate.**

**Dense training learning rate.** We start from existing scaling laws for batch size and learning rate. Prior work (Porian et al., 2024) suggests that the optimal learning rate decreases as model size increases. In (Porian et al., 2024), the DeepSeek fit is

$$\text{LR} = 0.17 \times N^{-0.25}, \tag{16}$$

where $N$ denotes the model size.

To verify this trend in our setting, we perform a learning-rate sweep on Llama models with different sizes under a fixed training budget of 2.6B tokens. The tested models include 120M, 240M, 480M, and 960M parameters. The results are shown in Table 2.

From these experiments, the optimal dense-training learning rates are approximately $2^{-9.75}, 2^{-10}, 2^{-10.5}$, and $2^{-10.75}$ for 120M, 240M, 480M, and 960M models, respectively. This confirms that the optimal learning rate decreases with model size and approximately follows

$$\text{LR} \propto 2^{-0.25 \log_2(N)}. \tag{17}$$

Therefore, we use the following dense-training learning-rate rule:

$$\text{LR}_{\text{dense}}(N) = 2^{-8.525} \times 2^{-0.25 \log_2(N)}. \tag{18}$$

Equivalently,

$$\text{LR}_{\text{dense}}(N) \propto N^{-0.25}. \tag{19}$$

**Sparse training learning rate.** Next, we examine whether the dense-training learning rate can be directly applied to dynamic sparse training. We conduct sparse training experiments on the Llama-1.92B model with sparsity levels $S = 0.75$ and $S = 0.875$. For dense training, the selected learning rate for Llama-1.92B is $2^{-11}$. We then compare this dense learning rate with scaled-up learning rates. The results are shown in Table 3.

*Table 3.* Learning-rate sweep for dynamic sparse training of LLaMA-1.92B on 5.4B tokens. Lower perplexity is better.

| Sparsity $(S)$ | Learning rate | Perplexity |
|---|---|---|
| 0.75 | $2^{-8.75}$ | 17.24 |
| 0.75 | $2^{-9} = 2^{-11} \times 4$ | **17.19** |
| 0.75 | $2^{-9.25}$ | 17.39 |
| 0.875 | $2^{-7.75}$ | 18.31 |
| 0.875 | $2^{-8} = 2^{-11} \times 8$ | **18.26** |
| 0.875 | $2^{-8.25}$ | 18.32 |

For sparsity $S = 0.75$, the active parameter ratio is $1 - S = 0.25$. Scaling the dense learning rate by the inverse active ratio gives

$$\frac{1}{1 - S} = 4, \tag{20}$$

and therefore

$$2^{-10.75} \times 4 = 2^{-8.75}. \tag{21}$$

This learning rate achieves the best perplexity of 18.15.

Similarly, for sparsity $S = 0.875$, the active parameter ratio is $1 - S = 0.125$. Scaling the dense learning rate by the inverse active ratio gives

$$\frac{1}{1 - S} = 8, \tag{22}$$

and therefore

$$2^{-10.75} \times 8 = 2^{-7.75}. \tag{23}$$

This learning rate achieves the best perplexity of 19.59.

These results show that the optimal learning rate from dense training cannot be directly transferred to dynamic sparse training. Instead, the learning rate should increase as the sparsity level increases. Empirically, the scaling factor is approximately the inverse active parameter ratio, i.e., $1/(1 - S)$.

Therefore, for dynamic sparse training, we use

$$\text{LR}_{\text{sparse}}(N, S) = \frac{\text{LR}_{\text{dense}}(N)}{1 - S}, \tag{24}$$

where

$$\text{LR}_{\text{dense}}(N) = 2^{-8.525} \times 2^{-0.25 \log_2(N)}. \tag{25}$$

Combining the two equations gives the final learning-rate rule:

$$\text{LR}_{\text{sparse}}(N, S) = \frac{2^{-8.525} \times 2^{-0.25 \log_2(N)}}{1 - S} \tag{26}$$

where $N$ is the model size and $S$ is the sparsity level.

## D. Fitting procedure

### D.1. Fitting on unique data without non-repeating

First, we use the validation loss values for both the dense and sparse models, evaluated on **unique, non-repeating data**, to fit the scaling law for the sparse and dense model:

$$L(N, D, S) = \frac{A\big((1 - S)^\epsilon + PS^\mu\big)}{N^\alpha} + \frac{B}{D^\beta} + E$$

Here:

- $N$ is the number of non-zero parameters,

- $D$ is the number of unique training tokens,

- $A, B, E, P, \epsilon, \mu, \alpha, \beta$ are fitting constants.

We fit and validate the scaling law using samples from dense and sparse models trained on unique, non-repeating data, spanning model sizes from 15M to 3.84B parameters, sparsity levels from 0.0 to 0.9375, and training tokens from 1.3B to 41.6B.

The fitted parameters are:

$$A, B, E, \alpha, \beta, \epsilon, \mu, P = [5.24073718, 6.10303089, 0.85355013, 0.32682177, 0.32682177, -0.00968044, 0.84661793, -0.27288932]$$

The resulting fit achieves a loss of $1.36 \times 10^{-3}$ and an $R^2$ score of 98.3%.

### D.1.1. OPTIMAL MODEL AND DATA SCALING UNDER SPARSITY

**Sparse FLOPs.** Under a compute budget that counts only non-zero operations, we assume

$$C = 6ND. \tag{27}$$

Substituting $D = C/(6N)$ into the loss gives

$$L(N, S) = A\big((1-S)^\varepsilon + PS^\mu\big)N^{-\alpha} + B6^\beta C^{-\beta}N^\beta + E. \tag{28}$$

Let

$$F(S) = (1-S)^\varepsilon + PS^\mu, \qquad G = \left(\frac{\alpha A}{\beta B}\right)^{\frac{1}{\alpha+\beta}}. \tag{29}$$

Minimizing with respect to $N$ yields the closed-form optima

$$N^*(S) = G(C/6)^{\frac{\beta}{\alpha+\beta}} F(S)^{\frac{1}{\alpha+\beta}}, \qquad D^*(S) = G^{-1}(C/6)^{\frac{\alpha}{\alpha+\beta}} F(S)^{-\frac{1}{\alpha+\beta}}, \tag{30}$$

which satisfy the sparse FLOPs constraint $N^*D^* = C/6$.

These expressions show that sparsity influences the optimal allocation of parameters and data only through the multiplicative factor $F(S)$:

$$N^* \propto F(S)^{\frac{1}{\alpha+\beta}}, \qquad D^* \propto F(S)^{-\frac{1}{\alpha+\beta}}. \tag{31}$$

Importantly, sparsity does not change the scaling exponents with respect to compute:

$$N^* \propto C^{\frac{\beta}{\alpha+\beta}}, \qquad D^* \propto C^{\frac{\alpha}{\alpha+\beta}}. \tag{32}$$

Sparsity only modifies the constant prefactors through $F(S)$, while the fundamental Chinchilla-style scaling with compute remains unchanged.

**Dense-FLOPs budget.** If compute is instead measured in dense FLOPs,

$$C = \frac{6ND}{1-S}, \tag{33}$$

then substituting $D = \frac{C(1-S)}{6N}$ into the loss and minimizing with respect to $N$ yields

$$N^* = G(C/6)^{\frac{\beta}{\alpha+\beta}}(1-S)^{\frac{\beta}{\alpha+\beta}} F(S)^{\frac{1}{\alpha+\beta}}, \tag{34}$$

$$D^* = G^{-1}(C/6)^{\frac{\alpha}{\alpha+\beta}}(1-S)^{-\frac{\alpha}{\alpha+\beta}}F(S)^{-\frac{1}{\alpha+\beta}}, \tag{35}$$

where

$$F(S) = (1-S)^\varepsilon + PS^\mu, \qquad G = \left(\frac{\alpha A}{\beta B}\right)^{\frac{1}{\alpha+\beta}}.$$

Letting $a = \beta/(\alpha+\beta)$ and $b = \alpha/(\alpha+\beta)$, we can write

$$N^* = G(C/6)^a(1-S)^a F(S)^{\frac{1}{\alpha+\beta}}, \qquad D^* = G^{-1}(C/6)^b(1-S)^{-b}F(S)^{-\frac{1}{\alpha+\beta}}. \tag{36}$$

When $S = 0$ (dense training), $F(S) = 1$, and these expressions reduce exactly to the standard Chinchilla scaling laws.

### D.2. Fitting repeating data

**Fitting dense training.** Secondly, we use the loss values for the dense model, evaluated on repeating data, to fit the parameters $R_D$ and $R_N$ in the scaling law for dense models with repeated data. In this step, the parameters $A, B, E, \alpha, \beta, \epsilon, \mu, P$ are fixed from the previous fitting stage:

$$A, B, E, \alpha, \beta, \epsilon, \mu, P = [5.24073718, 6.10303089, 0.85355013, 0.32682177, 0.32682177, -0.00968044, 0.84661793, -0.27288932]$$

The scaling law for dense training is given by:

$$L(N', D', S) = \frac{A\big((1-S)^\epsilon + PS^\mu\big)}{N'^\alpha} + \frac{B}{D'^\beta} + E$$

Here, the effective dataset size $D'$ and effective model size $N'$ are computed as:

$$D' = U_d + U_d \cdot R_d^* \left(1 - e^{-\frac{R_d}{R_d^*}}\right),$$

$$N' = U_n + U_n \cdot R_n^* \left(1 - e^{-\frac{R_n}{R_n^*}}\right),$$

where $U_d$ and $U_n$ represent the number of unique tokens and unique parameters, respectively, and $R_d$ and $R_n$ denote the number of repeated tokens and repeated parameters. We fit the parameter $R_d^*$ and $R_n^*$ for dense training with repetition up to 16 epochs.

The values of the parameters after fitting are

$$R_d^*, R_n^* = [11.08763712, 4.40474882].$$

The resulting fit achieves a loss of $5.03 \times 10^{-3}$ and an $R^2$ score of 98.2%.

**Fitting sparse training.** Then, we use the loss values for the sparse model, evaluated on repeating data, to fit the parameters $\lambda_1$, $\sigma_1$, $\lambda_2$ and $\sigma_2$ in the scaling law for sparse models with repeated data. In this step, the parameters $A, B, E, \epsilon, \alpha, \beta, R_D^*, R_N^*$ are fixed from the previous fitting stage:

$$A, B, E, \alpha, \beta, \epsilon, \mu, P, R_d^*, R_n^*$$
$$= [5.24073718, 6.10303089, 0.85355013, 0.32682177, 0.32682177,$$
$$0.00968044, 0.84661793, 0.27288932, 11.08763712, 4.40474882]$$

The scaling law is defined as:

$$L(N', D', S) = \frac{A\big((1-S)^\epsilon + PS^\mu\big)}{N'^\alpha} + \frac{B}{D'^\beta} + E$$

where the effective dataset size $D'$ and effective model size $N'$ are given by:

$$D' = U_d + U_d \cdot R_d^*(S)\left(1 - e^{-\frac{R_d}{R_d^*(S)}}\right),$$

$$N' = U_n + U_n \cdot R_n^*(S)\left(1 - e^{-\frac{R_n}{R_n^*(S)}}\right),$$

$$R_d^*(S) = R_d^*\left(1 + \lambda_1 S + \sigma_1 S^2\right), R_n^*(S) = R_n^*\left(1 + \lambda_2 S + \sigma_2 S^2\right),$$

we then fit $\lambda_1, \sigma_1, \lambda_2$ and $\sigma_2$ the results from models with sparsity levels from 0.0 (dense) to 0.975, with training epoch up to 16. The values of the parameters after fitting are

$$\lambda_1, \sigma_1, \lambda_2, \sigma_2 = [1.82159486, 1.36557887, 1.90420893, 2.79936732]$$

The resulting fit achieves a loss of $1.15 \times 10^{-3}$ and an $R^2$ score of 98.2%.

### D.3. Formulations Explanation

#### D.3.1. DEPENDENCE OF THE LOSS GAP ON THE AMOUNT OF UNIQUE DATA

We analyze how the loss gap between training on unique and repeated data scales with the amount of unique data $U_d$ under the previous scaling law (Muennighoff et al., 2023).

We consider the standard data-dependent term of the scaling law,

$$L(N, D) = \frac{A}{N^\alpha} + \frac{B}{D^\beta} + E, \tag{37}$$

and define the loss gap as

$$\Delta L \equiv L_{\text{rep}} - L_{\text{uniq}}. \tag{38}$$

**Unique data.** When training on unique data, the total number of training tokens is

$$D_{\text{uniq}} = U_d(1 + R_D), \tag{39}$$

where $R_D$ denotes the number of data repetitions.

**Repeated data.** Under repeated training, the effective dataset size is given by

$$D'_{\text{rep}} = U_d\left[1 + R_D^*\left(1 - e^{-R_D/R_D^*}\right)\right], \tag{40}$$

where $R_D^*$ denotes the saturation scale controlling the diminishing returns from repeated exposure.

**Loss gap.** Substituting $D_{\text{uniq}}$ and $D'_{\text{rep}}$ into the data-dependent term yields

$$\Delta L = B\left(\frac{1}{(D'_{\text{rep}})^\beta} - \frac{1}{(D_{\text{uniq}})^\beta}\right)$$
$$= BU_d^{-\beta}\left[\left(1 + R_D^*\left(1 - e^{-R_D/R_D^*}\right)\right)^{-\beta} - (1 + R_D)^{-\beta}\right]. \tag{41}$$

Since

$$1 + R_D^*\left(1 - e^{-R_D/R_D^*}\right) < 1 + R_D \quad \text{for any } R_D > 0, \tag{42}$$

the bracketed term is strictly positive and independent of $U_d$. Therefore, the loss gap satisfies

$$\Delta L \propto U_d^{-\beta} \tag{43}$$

and decreases monotonically as the amount of unique data increases.

### D.3.2. DEPENDENCE OF THE LOSS GAP ON PARAMETER REPETITION

We analyze how the loss gap between training on unique and repeated data scales with the model size under the previous scaling law (Muennighoff et al., 2023). Under fixed $U_d$ and $U_n$, increasing model size is equivalent to increasing the parameter repetition ratio $R_N$. Since the model size satisfies $N = U_n(1 + R_N)$, analyzing the dependence of the loss gap on $R_N$ is equivalent to analyzing its dependence on the model size $N$.

Throughout this analysis, we fix the data-side quantities $U_d$ and $R_D$, such that the effective dataset sizes

$$D'_{\text{uniq}} = U_d(1 + R_D), \qquad D'_{\text{rep}} = U_d + U_d R_D^* \left(1 - e^{-R_D/R_D^*}\right) \tag{44}$$

are constants independent of $R_N$.

**Effective parameters.** The effective number of parameters under repeated training is given by

$$N'_{\text{rep}} = U_n + U_n R_N^* \left(1 - e^{-R_N/R_N^*}\right), \tag{45}$$

where $R_N^*$ denotes the saturation scale for parameter repetition. Under unique training, the model architecture is unchanged and parameters do not undergo repetition saturation. Therefore, the effective number of parameters is

$$N'_{\text{uniq}} = U_n(1 + R_N). \tag{46}$$

**Loss gap.** We define the loss gap as

$$\Delta L(R_N) \equiv L_{\text{rep}} - L_{\text{uniq}}. \tag{47}$$

Substituting the expressions above yields

$$\Delta L(R_N) = \left[\frac{A}{(N'_{\text{rep}})^\alpha} + \frac{B}{(D'_{\text{rep}})^\beta}\right] - \left[\frac{A}{(N'_{\text{uniq}})^\alpha} + \frac{B}{(D'_{\text{uniq}})^\beta}\right]$$
$$= A\left(\frac{1}{(N'_{\text{rep}})^\alpha} - \frac{1}{\left(U_n(1 + R_N)\right)^\alpha}\right) + B\left(\frac{1}{(D'_{\text{rep}})^\beta} - \frac{1}{(D'_{\text{uniq}})^\beta}\right). \tag{48}$$

As the parameter repetition ratio $R_N$ increases, the effective number of parameters under repeated training, $N'_{\text{rep}}$, grows increasingly slowly and eventually saturates. In contrast, under unique training the effective parameter count $N'_{\text{uniq}} = U_n(1 + R_N)$ continues to grow linearly with $R_N$. As a result, the parameter-dependent term in Eq. 48 becomes increasingly dominated by the second term, leading to a widening loss gap between repeated and unique training as $R_N$ increases.

### D.3.3. THE RELATIONSHIP OF THE LOSS GAP AND $R_d^*$

**Proposition D.1** (Effect of the data saturation scale). *Let $U_d > 0$, $R_d > 0$, $R_d^* > 0$, $B > 0$, and $\beta > 0$. Define the effective data under repetition as*

$$D'(R_d^*) = U_d + U_d R_d^* \left(1 - e^{-R_d/R_d^*}\right), \tag{49}$$

*and the corresponding data-dependent loss term as*

$$L_{\text{rep}}(R_d^*) = \frac{B}{D'(R_d^*)^\beta}. \tag{50}$$

*Let the ideal fully-unique-data baseline be*

$$L_{\text{uniq}} = \frac{B}{\left(U_d(1 + R_d)\right)^\beta}. \tag{51}$$

*Then, for fixed $R_d > 0$, the gap*

$$G(R_d^*) = L_{\text{rep}}(R_d^*) - L_{\text{uniq}} \tag{52}$$

*is strictly decreasing in $R_d^*$.*

*Proof.* Fix $R_d > 0$ and let $t = R_d^* > 0$. Define

$$D'(t) = U_d + U_d\, t \left(1 - e^{-R_d/t}\right).$$

We first show that $D'(t)$ is strictly increasing in $t$. Consider

$$\phi(t) = t \left(1 - e^{-R_d/t}\right).$$

Differentiating,

$$\phi'(t) = 1 - e^{-R_d/t} \left(1 + \frac{R_d}{t}\right).$$

Let $x = R_d/t > 0$. Then

$$\phi'(t) = 1 - e^{-x}(1 + x).$$

Since $e^{-x}(1 + x) < 1$ for all $x > 0$, we have $\phi'(t) > 0$, so $D'(t)$ is strictly increasing in $t$.

Next, since $\beta > 0$, the function $h(z) = B/z^\beta$ is strictly decreasing for $z > 0$. Therefore

$$L_{\text{rep}}(t) = \frac{B}{D'(t)^\beta}$$

is strictly decreasing in $t$.

Finally, $L_{\text{uniq}} = B/(U_d(1 + R_d))^\beta$ is constant with respect to $t$. Hence

$$G(t) = L_{\text{rep}}(t) - L_{\text{uniq}}$$

is strictly decreasing in $t = R_d^*$.

## E. Sensitivity Analysis

To assess the robustness of the optimal-sparsity predictions, we perform a leave-one-density sensitivity analysis. Specifically, we repeatedly refit the scaling law after removing all training runs corresponding to one sparsity level from the fitting set. For each refitted model, we recompute the Iso-FLOPs curves, the dense reference optimum, the compute-optimal sparse operating point, and the loss-optimal sparsity. This produces a collection of fitted predictions, each corresponding to a slightly different training subset. We then summarize these predictions by plotting the mean loss curve across refits and using shaded bands to indicate the standard deviation across refitted curves.

In the figure, the uncertainty bands are computed from mean-centered loss curves. That is, for each refit we subtract the average loss of the corresponding curve before computing the standard deviation across refits. This removes global vertical offsets between fitted models and emphasizes uncertainty in the *shape* of the sparsity–loss curve. As a result, the bands should be interpreted as shape sensitivity rather than absolute prediction uncertainty. This distinction is important because different refits may shift the overall loss level while preserving a similar sparsity-dependent trend.

The sensitivity results show that the predicted curves are most stable in the moderate-sparsity region, where both the compute-optimal and loss-optimal operating points are located. In contrast, the uncertainty bands become wider near the boundaries, especially close to dense training and extreme sparsity. This behavior is expected: boundary regions are more sensitive to the removal of individual sparsity levels and are more strongly affected by the curvature terms in the fitted sparsity-dependent functions. Importantly, despite the increased sensitivity near the boundaries, the main conclusion remains stable across refits: moderate sparsity consistently provides the best trade-off between data reuse and model capacity, while very high sparsity leads to diminishing returns. The compute-saving trend is also robust: across refits and data budgets, the compute-optimal sparse operating point matches the dense reference loss with roughly 7–11× fewer training FLOPs, indicating that the observed efficiency gain is not an artifact of a single fitted model.

## F. Limitations

To our knowledge, this is the first systematic empirical study of dynamic sparse training (DST) under data-constrained scaling. Nevertheless, several limitations remain.

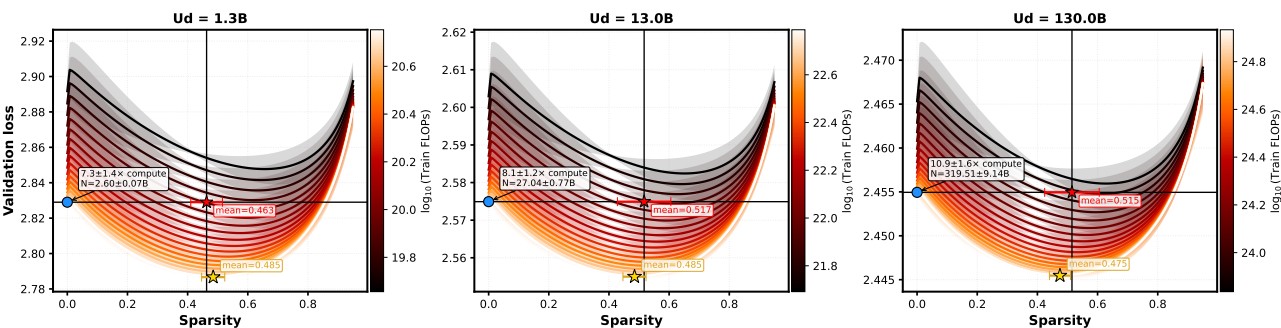

*Figure 7.* Optimal sparsity under fixed unique-data budgets of $U_d = 1.3B$, 13B, and 130B tokens. Curves show validation loss across sparsity along Iso-FLOPs contours, with color indicating $\log_{10}$ training FLOPs and shaded bands showing leave-one-density sensitivity. The blue point is the dense reference optimum, the red star is the compute-optimal sparse configuration matching dense performance, and the yellow star is the loss-optimal sparse configuration. Moderate sparsity consistently yields dense-equivalent performance with fewer FLOPs.

**Scale and Data Diversity.** Our experiments span a broad range of sparsity levels, model sizes, and data budgets, but the largest models remain below the scale of frontier LLMs. Additionally, we focus primarily on the C4 dataset. Different domains (e.g., code or multilingual data) may exhibit different repetition dynamics, which could shift quantitative saturation behavior. Extending the study to larger model scales and more diverse data sources would require substantially greater computational resources and is therefore left for future work.

**System and Practical Efficiency.** Our compute analysis distinguishes between sparse FLOPs and dense-equivalent FLOPs. In practice, hardware and system inefficiencies may reduce the realizable speedups from sparsity. Bridging the gap between algorithmic and hardware-efficient sparsity remains an open challenge.

**Theoretical explanation.** The proposed sparsity-aware scaling law is phenomenological and chosen for empirical fit and interpretability. While it captures observed trends well, it does not yet provide a theoretical explanation for how sparsity alters effective data or parameter saturation.

Overall, our work provides an initial characterization of sparse data-constrained scaling, and we hope it motivates further theoretical, empirical, and systems-level investigation.

