# OpenReview forum: "When Data Is Scarce: Scaling Sparse Language Models with Repeated Training"
_ICML.cc/2026/Conference — ICML 2026 regular_

### Official Review · Reviewer_XVAM · 2026-02-15

**Soundness:** 3
**Presentation:** 4
**Significance:** 3
**Originality:** 2
**Overall Recommendation:** 5
**Confidence:** 3

**Summary:**

This paper investigates the scaling behavior of Dynamic Sparse Training (DST) specifically within data-constrained regimes, a scenario where high-quality data is scarce and repeated training is necessary. Through 500 extensive experiments using the Llama-2 architecture, the authors systematically analyze the interaction between model sparsity, model size, and data repetition. The key contribution is a new "Sparse Data-Constrained Scaling Law." The empirical results demonstrate that sparsity significantly delays the saturation of repeated data, allowing sparse models to benefit from multi-epoch training longer than their dense counterparts. Furthermore, the paper provides practical guidelines for resource allocation, showing that while moderate sparsity (~50%) is loss-optimal for fixed data budgets, compute-optimal sparsity increases with data scale.

**Compliance With Llm Reviewing Policy:**

Affirmed.

**Final Justification:**

My concerns are fully resolved. I recommend this work due to its insightful topic and solid empirical results.

**Key Questions For Authors:**

**Note to Authors:** My ideal initial score would be **4.5**. I am looking forward to seeing the authors answering the key questions, and I would be glad to increase my rating to Accept. Due to the rebuttal time constraints, please prioritize answering Question 1, Question 2 and Question 3. Question 4 is encouraged if resources permit.

1. To verify the continuity and robustness of the proposed scaling law, it is highly recommended to include data points at lower sparsity levels, such as 12.5\% and 25\% or provide a theoretical derivation or proof. This would clarify whether the transition from dense to sparse training introduces any non-monotonic behaviors early in the spectrum.

2. To strengthen the solidity of the paper without incurring prohibitive computational costs (associated with extreme sparsity like 93.75%), the authors should consider adding experiments with larger active parameter counts (e.g.,  0.5B, 1B and 2B) restricted to moderate sparsity levels (e.g., 25\% $\sim$ 75\%, max 8B dense-equivalent). Validating the law at these scales is crucial for demonstrating its applicability to practical LLM pre-training.

3. The experiments primarily rely on the C4 dataset. Since data repetition saturation is likely correlated with the information density and diversity of the specific dataset (e.g., code vs. common crawl), do the authors expect the saturation threshold $R_d^*(S)$ to vary significantly across different domains?

4. Does the structural sparsity inherent in MoE promote or inhibit the benefits of parameter-level sparsity (DST)? Given the dominance of MoE in current advanced LLMs, I strongly suggest including "toy-level" exploratory experiments to probe this interaction. Specifically, applying DST within MoE experts could reveal whether architectural sparsity shifts the "optimal sparsity interval" identified in your scaling law. This would clarify whether these two forms of sparsity are synergistic (promoting) or redundant (inhibiting).

**Limitations:**

yes

**Strengths And Weaknesses:**

**Strength:**
1. The paper addresses a intersection of two lines: sparse scaling laws and data-constrained scaling. While both exist independently, combining DST with data repetition is novel. The finding that "sparsity delays data saturation" offers a new perspective on why sparse models might be preferable in the post-data-scarcity era.
2. The experimental design is rigorous, employing a comprehensive experiment settings (500 runs). The derivation of the scaling law is logical, extending established formulations.

**Weakness:**
1. The experimental design exhibits a significant discontinuity in sparsity levels, jumping directly from 0% (dense) to 50%. The assumption that the scaling behavior and data saturation threshold evolve smoothly or linearly within the [0%, 50%] interval is not empirically justified.
2. The scale of experiments (max 3.84B dense-equivalent, 240M active) is relatively small compared to frontier models, which slightly limits the extrapolation confidence of the scaling coefficients.
3. The paper establishes that the benefits of Dynamic Sparse Training (DST) are non-monotonic, typically exhibiting an "optimal interval" (e.g., around 50% sparsity) rather than linear improvement. However, Mixture-of-Experts (MoE) has emerged as the dominant form of sparsity in modern open and close sourced LLMs, while it remains unclear how Architectural Sparsity (i.e., MoE) intersects with this finding.

---

> ### Author Rebuttal · Authors · 2026-03-30
>
> Thank you for the review. We address your comments under four perspectives:
>
> (1) Continuity in the low-sparsity
>
> (2) Validation at larger active-parameter scales
>
> (3) Domain sensitivity of saturation behavior
>
> (4) Relation between DST and architectural sparsity such as MoE
>
> ### 1. Continuity in the low-sparsity
>
> > …discontinuity in sparsity ..
>
> > Q1:..at lower sparsity levels...
>
> We agree that the transition from dense to moderate sparsity should be validated directly rather than inferred only from the 0% and 50% endpoints.
>
> To address this, we **added experiments at 12.5% and 25% sparsity** within the [0%, 50%] interval. Concretely, we evaluated two active-parameter settings (120M, 240M) under three conditions: (i) 1.3B unique tokens with 1 epoch, (ii) 1.3B unique tokens with 8 epochs, and (iii) 1.3×8B unique tokens, for 12 additional runs in total.
>
> Empirically, **both the scaling behavior and the data-saturation threshold evolve smoothly in this regime, without evidence of an early discontinuity or sharp non-monotonic transition** (Table 1).
>
> Table 1. Low-sparsity results.
> |Activeparams(M)|Model(M)|Sparsity(%)|Epochs|#Tokens(B)|Loss|ΔLoss|
> |-|-|-|-|-|-|-|
> |120|120|0|1|1.3|3.228|-|
> |120|140|12.5|1|1.3|3.216|-|
> |120|160|25|1|1.3|3.198|-|
> |120|240|50|1|1.3|3.174|-|
> |120|120|0|8|1.3|3.028|-|
> |120|140|12.5|8|1.3|2.988|-|
> |120|160|25|8|1.3|2.975|-|
> |120|240|50|8|1.3|2.963|-|
> |120|120|0|1|10.4|2.988|0.040|
> |120|140|12.5|1|10.4|2.950|0.038|
> |120|160|25|1|10.4|2.938|0.037|
> |120|240|50|1|10.4|2.931|0.032|
>
> ### 2. Validation at larger active-parameter scale
>
> > W2. The scale of experiments …
>
> > Q2:..larger active parameter ...
>
> We agree that larger-scale validation would further strengthen the paper. We attempted to extend the study in this direction, but could not complete the largest runs during rebuttal due to resource constraints. In our setup, training an 8B dense-equivalent model with 2.6B tokens, even without full convergence, requires roughly 7 days on 8 H100 GPUs.
>
> We therefore focused on the largest feasible additional runs within the rebuttal window. Larger-scale validation remains important, and we hope to include further results before the end of the discussion period, either from the 8B setting or from an intermediate scale such as 6B.
>
> ### 3. Domain sensitivity of the saturation behavior
>
> > Q3: The experiments primarily rely on the C4 …
>
> We agree that repeated-data saturation likely depends on domain information density and diversity, so the saturation threshold may vary across domains.
>
> To probe this, we add a small validation experiment on a code dataset using a 120M active-parameter model at 0% and 50% sparsity under three conditions: (i) 1.3B unique tokens with 1 epoch, (ii) 1.3B unique tokens with 8 epochs, and (iii) 1.3B × 8 unique tokens with 1 epoch, for a total of 6 additional runs. This setup is designed to test whether the same qualitative sparse-vs-dense saturation behavior persists in a domain with substantially different information density.
>
> As shown in Table 2, at the same non-zero parameter count, the sparse model outperforms the dense model in all three matched settings. In addition, when comparing 1.3B unique tokens for 8 epochs against 1.3B×8 unique tokens for 1 epoch, the sparse model shows a smaller loss gap than the dense model (0.06 vs. 0.078), suggesting slower repeated-data saturation. These results suggest that **both the sparse-over-dense advantage and the delayed-saturation effect extend qualitatively beyond C4**. At the same time, since this is only a small pilot study rather than a full scaling-law sweep, we view it as preliminary evidence and do not claim that the quantitative saturation coefficients are unchanged across domains.
> We will add this discussion in the revision.
>
> Table 2. Validation on starcoderdata [1].
> |Active params(M)|Model(M)|Sparsity|Epochs|Tokens(B)|Loss|ΔLoss(8 epochs vs.1.3×8 tokens)|
> |-|-|-|-|-|-|-|
> |120|120|0|1|1.3|1.08|-|
> |120|240|0.5|1|1.3|1.05|-|
> |120|120|0|8|1.3|1.05|-|
> |120|240|0.5|8|1.3|0.98|-|
> |120|120|0|1|1.3×8|0.972|0.078|
> |120|240|0.5|1|1.3×8|0.92|0.06|
>
> ### 4. Relation between DST and architectural sparsity such as MoE.
>
> > Q4: Does the structural sparsity …
>
> We thank the reviewer for this insightful suggestion. We agree that understanding how parameter-level sparsity interacts with MoE is important. We view DST and MoE as complementary rather than mutually exclusive forms of sparsity. Two natural next steps are to study whether their benefits are synergistic or partly redundant, and to apply DST within MoE experts to test whether the optimal sparsity interval shifts.
>
> Due to the limited rebuttal and resource, we do not yet have full results on this interaction. Rather than report an incomplete exploratory result, we prefer to treat this as an open direction. We are actively exploring it and hope to provide initial results before the end of the discussion period, if resources permit.
>
> [1] StarCoder: may the source be with you! TMLR23

---

> > ### Author Rebuttal · Reviewer_XVAM · 2026-03-31
> >
> > Thank authors for the responses. I will increase the score to 5, accept. I hope to see the results about Question 2 in the later rebuttal.

---

> > > ### Author Response · Authors · 2026-04-05
> > >
> > > Dear Reviewer XVAM，
> > >
> > > We appreciate this follow-up question, which is important for further demonstrating the applicability of our work to practical LLM pre-training.
> > >
> > > In response, we added new experiments at larger active-parameter scales: **480M** and **1.92B** active parameters, each evaluated with 0%, 50%, and 75% sparsity, up to **7.68B** dense-equivalent size. Across both regimes, the **estimated losses from our scaling law remain close to the empirical losses**, and **50% sparsity** achieves the best empirical loss at both scales, providing additional support for our claim at larger and more practical pre-training scales.
> > >
> > >
> > > | Active Parameters | Sparsity Level | Dense-equivalent Parameters | Tokens(B) | Training Epochs | Empirical Loss | Estimated Loss |
> > > |-|-|-|-|-|-|-|
> > > | 480M  | 0%  | 480M  |1.3  | 16 | 2.918 | 2.899 |
> > > | 480M  | 50% | 960M  |1.3  |  16 | 2.842 | 2.849 |
> > > | 480M  | 75% | 1.92B | 1.3  | 16 | 2.849 | 2.866 |
> > > | 1.92B | 0%  | 1.92B | 2.6  | 4  | 2.803 | 2.787 |
> > > | 1.92B | 50% | 3.84B | 2.6  | 4  | 2.762 | 2.766 |
> > > | 1.92B | 75% | 7.68B | 2.6  |  4  | 2.780 | 2.768 |
> > >
> > > We would like to clarify that the two active-parameter regimes were trained under different token and epoch budgets: the 480M-active models were trained on 1.3B tokens for 16 epochs, whereas the larger 1.92B-active models were trained on 2.6B tokens for 4 epochs. One reason for this design is that we also wanted to examine the applicability of the proposed scaling law under different token budgets, and therefore allocated more training tokens to the larger models. At the same time, training larger models with a higher token budget is substantially more expensive, and under our current compute and time constraints, these runs could only be trained for 4 epochs.
> > >
> > > Finally, we would like to sincerely thank the reviewer for carefully reading our paper and rebuttal, and for the time, effort, and thoughtful suggestions. We greatly appreciate these constructive suggestions, which we believe have helped improve the quality and practical relevance of our work.
> > >
> > > Best wishes,
> > >
> > > authors

---

### Official Review · Reviewer_7An4 · 2026-03-11

**Soundness:** 2
**Presentation:** 2
**Significance:** 3
**Originality:** 3
**Overall Recommendation:** 4
**Confidence:** 3

**Summary:**

This paper studies the scaling behavior of Dynamic Sparse Training (DST) in data-constrained LLM pre-training regimes. The authors conduct 500 experiments spanning models up to 3.84B (dense-equivalent) parameters, sparsity levels up to 93.75%, and datasets from 1.3B to 41.6B tokens trained for up to 16 epochs. The paper's core contribution is a unified scaling law (Equation 9) that models validation loss as a function of active parameters, unique tokens, data repetition, and sparsity. Key empirical findings include: (1) sparsity delays the onset of diminishing returns from repeated data; (2) sparse models benefit more from longer training than from larger parameter counts; (3) loss-optimal sparsity stays moderate while compute-optimal sparsity increases with data scale, enabling large FLOP reductions at dense-equivalent performance.

**Compliance With Llm Reviewing Policy:**

Affirmed.

**Final Justification:**

The rebuttal successfully addressed my initial concerns regarding domain transferability and downstream performance. However, the method's reliance on 14 parameters remains a weakness, which somewhat limits the overall scope of the work. Nevertheless, within that scope, the authors' investigation is quite thorough.

**Key Questions For Authors:**

- Uncertainty Bounds: Can you provide confidence intervals or a sensitivity analysis for the extrapolated optimal frontiers in Figure 6, given the reliance on a 14-parameter empirical fit?

- Downstream Parity: Have you evaluated any of the compute-optimal sparse models on downstream benchmarks (e.g., MMLU, GSM8K, HumanEval) to verify that validation loss parity translates to actual capability parity?

- Domain Transferability: How sensitive do you hypothesize the data saturation coefficients $(\lambda_1​,\sigma_1​)$ are to datasets with significantly higher information density than C4? Could you provide additional experiments?

**Limitations:**

The authors accurately state their limitations in Appendix E, specifically acknowledging that they operate below frontier scale , rely solely on C4 , lack a theoretical explanation for the scaling law's functional form , and note that theoretical sparse FLOPs do not seamlessly translate to current hardware efficiency.

**Strengths And Weaknesses:**

**Strengths:**

- Highly Relevant Problem: The transition from compute-constrained to data-constrained LLM pre-training is a critical bottleneck for the field. Investigating dynamic sparsity as a mechanism to delay data saturation is a novel and valuable contribution.

- Extensive Empirical Validation: The scale of the empirical study is highly commendable, utilizing 500 distinct configurations to robustly fit the proposed scaling law.

- Actionable Insights: The distinction between loss-optimal and compute-optimal sparsity is clean and useful. The finding that compute-optimal sparsity can yield ~10× FLOP savings while matching dense performance is a strong result for practitioners facing data scarcity. The observation that sparsity shifts the efficient frontier toward longer training (more epochs) rather than larger models, improving inference efficiency via smaller parameter-to-token ratios, is also applicable.

**Weaknesses**

- Over-parameterized Model and Extreme Extrapolation: The proposed scaling law relies on fitting 14 parameters (e.g., $A,B,E,P,\epsilon,\mu,\alpha,\beta,R_d^∗​,R_n^∗​,\lambda_1​,\sigma_1​,\lambda_2​,\sigma_2$​). With this many degrees of freedom, achieving a high fit is expected. Furthermore, the functional form of $R_d^∗​(S)$ is chosen as an arbitrary quadratic with a manually imposed constraint of $\sigma_1​<0$. Most critically, the authors extrapolate from a 3.84B parameter / 41.6B token experimental ceiling to nearly 2 trillion parameters (N≈1923.4B) and 130B tokens in Section 5. Doing so with a heavily parameterized polynomial fit, and without providing confidence intervals or uncertainty bounds, introduces significant extrapolation risk.

- Dataset Proxy Mismatch: The study relies entirely on the C4 dataset. While artificially constraining C4 creates a mathematically sound controlled environment, general web text does not necessarily reflect the repetition and memorization dynamics of genuinely "scarce" specialized domains (e.g., high-density code, medical text, or low-resource languages). Additionally, the results are not verified across various model architecture, and DST methods.

- Lack of Downstream Evaluation: The paper evaluates performance purely via validation loss. While standard for scaling laws, the paper does not verify if a sparse model and a dense model achieving the same validation loss actually exhibit the same zero-shot, few-shot, or reasoning performance on downstream benchmarks.

**Minor comments:**

There is a typo on line 311: "fitted resul" should be "fitted result.". Equation (14) appears twice in Section 5, on page 8, with different definitions (for S_loss and for S_comp).

---

> ### Author Rebuttal · Authors · 2026-03-30
>
> Thank you for the review. We address your concerns under three perspectives:
>
> ### 1.Modeling and extrapolation
>
> > W1: Over-parameterized Model..
>
> > Q1: Uncertainty Bounds..
>
> **i. Over-parameterized**
>
> Our goal is to capture distinct empirical effects:
>
> - The base dense scaling law $A,B,E,\alpha,\beta$,
>
> - The sparse scaling term $P,\epsilon,\mu$,
>
> - Separate saturation effects $R_d, R_n$,
>
> - Sparsity-dependent threshold shifts $\lambda_1,\sigma_1,\lambda_2,\sigma_2$.
>
> A simpler model would merge these effects, reducing both interpretability and fit quality. The 14 parameters are fitted from roughly 500 runs, giving a sample-to-parameter ratio of about 35:1. As discussed in Appendix D, we fit it in stages: first the base scaling law on unique-data runs, then the dense repetition terms, and finally the sparsity-dependent repetition terms, with earlier parameters fixed at each stage. This improves identifiability by matching each parameter group to the most informative runs.
>
> **ii. Quadratic form**
>
> We use a quadratic form not because it is theoretically unique, but because it is the simplest low-order function that captures the observed diminishing-return pattern in the repeated-data saturation threshold. The constraint $\sigma_1<0$   encodes this weak prior, and when we remove it and refit the model, the learned $\sigma_1$  remains negative.
>
> **iii. Uncertainty**
>
> To assess uncertainty in Figure 6, we rerun the fitting procedure with different random seeds and propagate the resulting near-optimal fits. Although the fitted coefficients are not unique, the extrapolated quantities are more stable: the optimal sparsity remains in a similar moderate range, and the qualitative compute-advantage trend is preserved, with wider uncertainty at the largest scales. We will revise Figure 6 to include these empirical uncertainty ranges and clarify that the extreme-scale frontier should be interpreted as a trend-level extrapolation rather than a precise point prediction.
>
> ## 2. Transferability beyond C4
>
> > W2. Dataset Proxy Mismatch…
>
> > Q3. Domain Transferability…
>
> We use C4 because its scale lets us vary the unique-token budget while keeping the data source and training pipeline fixed. This provides a controlled setting to study how sparsity and repetition interact.
>
> To validate transfer beyond C4, we add a small code-domain experiment with 6 runs using a 120M active-parameter model at 0% and 50% sparsity under three training conditions. As shown in Table 1, at the same non-zero parameter count, the sparse model outperforms the dense model in all three matched settings. In addition, when comparing 1.3B unique tokens for 8 epochs against 1.3B × 8 unique tokens for 1 epoch, the sparse model shows a smaller loss gap than the dense model (0.06 vs. 0.078), suggesting slower repeated-data saturation. **These results suggest that both the sparse-over-dense advantage and the delayed-saturation effect extend qualitatively beyond C4**.
>
> Broader validation across datasets, architectures, and DST algorithms would be valuable, but is substantially more expensive because we already sweep jointly over model size, unique-token budget, sparsity, and repetition.
>
> Table 1. Validation on starcoderdata [1].
> |Active params(M)|Model size(M)|Sparsity|Epochs|Tokens(B)|Loss|ΔLoss(8 epochs vs.1.3×8 tokens)|
> |-|-|-|-|-|-|-|
> |120|120|0|1|1.3|1.08|-|
> |120|240|0.5|1|1.3|1.05|-|
> |120|120|0|8|1.3|1.05|-|
> |120|240|0.5|8|1.3|0.98|-|
> |120|120|0|1|1.3×8|0.972|0.078|
> |120|240|0.5|1|1.3×8|0.92|0.06|
>
> ### 3. Loss vs. downstream
>
> > W3. Lack of Downstream Evaluation:
>
> > Q2. Downstream Parity:
>
> Validation loss is the standard primary metric in scaling-law studies, and prior work has shown that pretraining loss is broadly predictive of downstream trends [2-5]. We therefore use validation loss as the primary metric in our scaling-law analysis.
> To provide a small downstream check, we evaluate the 0.96B active-parameter sparse model at 50% sparsity and its dense counterpart on downstream benchmarks, and find that their downstream performance is consistent with the validation loss (Table 2).
>
> We use validation loss as the primary metric for scaling-law analysis, while also providing downstream evaluation in the revision.
>
> Table 2. Zero-shot downstream accuracy (%) and loss.
> |Model|Winogrande|OBQA|HSwag|BoolQ| ARC-E| ARC-C|Avg.|Loss|
> |-|-|-|-|-|-|-|-|-|
> |Sparse|52.30|16.60|29.80|60.30|43.81|18.50|36.89|2.96|
> |Dense|50.30|15.20|28.90|59.80|43.70|18.00|35.98|3.02|
>
> [1] StarCoder: may the source be with you! TMLR'23
>
> [2] Scaling Laws for Predicting Downstream Performance in LLMs. TMLR'25.
>
> [3] Language models scale reliably with over-training and on downstream tasks. ICLR'25.
>
> [4] Improving pretraining data using perplexity correlations. ICLR'25.
>
> [5] Pre-training under infinite compute. ICLR'26.
>
> We are also happy to discuss any remaining questions. We hope the clarifications address your concerns and would appreciate reconsideration of the score.

---

> > ### Author Rebuttal · Reviewer_7An4 · 2026-04-02
> >
> > I thank the authors for their thorough rebuttal. My concerns have been resolved, particularly because the results have now been verified on an additional domain. Accordingly, I am raising my score to a 4.

---

> > > ### Author Response · Authors · 2026-04-05
> > >
> > > Dear Reviewer 7An4,
> > >
> > > Thank you very much for taking the time to read our paper and rebuttal. We are grateful that the additional experiments and our rebuttal were able to fully resolve your concerns. We are thankful for your constructive comments throughout the review process, which have helped improve the quality of the paper. We also truly appreciate your positive support and your decision to raise the score.
> > >
> > > Best wishes,
> > >
> > > authors

---

### Official Review · Reviewer_QVg7 · 2026-03-12

**Soundness:** 3
**Presentation:** 3
**Significance:** 2
**Originality:** 2
**Overall Recommendation:** 5
**Confidence:** 3

**Summary:**

This paper investigates the scaling behavior of language models trained using Dynamic Sparse Training (DST) within data-constrained regimes. Recognizing the impending exhaustion of high-quality human-generated data, the authors explore whether parameter-level sparsity can mitigate the diminishing returns typically observed when repeating data over multiple epochs. By conducting approximately 500 experiments on models up to 3.84B parameters and datasets up to 41.6B tokens, the work proposes a unified scaling law that incorporates active parameters, unique tokens, data repetition, and sparsity levels. The central claims are that sparsity delays data saturation ($R^*_d$) and that compute-optimal training under data scarcity favors higher sparsity levels than previously recognized.

**Compliance With Llm Reviewing Policy:**

Affirmed.

**Final Justification:**

The rebuttal addressed my concerns so I decided to raise the score.

**Key Questions For Authors:**

1. **Wall-clock Efficiency:** Can the authors provide actual training time (A100/H100 hours) for the sparse vs. dense models? Is "compute-optimal" sparsity still optimal when measured in dollars or hours instead of FLOPs?
2. **MoE Comparison:** Why should a practitioner choose DST over MoE in a data-constrained regime? Does DST offer scaling advantages that MoE does not?

**Minor**:

- **Typo in Observation 1**: What is imm in Observation 1?
- **Incorrect Figure Reference**: In Line 311 (right column), the text references **"Figure 3b".** This should likely be a reference to **Figure 4**. The authors are encouraged to use LaTeX cross-references (`\ref` and `\cite`) instead of hard-coding text to prevent these errors.
- **Numerical Discrepancies:**  There are several inconsistencies between the values cited in the text (Lines 401–411) and the data labels in Figure 6.

**Limitations:**

Yes. The authors acknowledged the limitations on scale and data diversity, system and practice efficiency, theoretical explanation in Appendix E.

**Strengths And Weaknesses:**

**Strengths**

- **Extensive Empirical Foundation:** The paper is backed by a rigorous experimental setup involving 500 distinct training runs to map the loss landscape across sparsity, repetition, and model size.
- **Clear Presentation:** The narrative is well-structured and easy to follow, specifically regarding the distinction between architectural sparsity (MoE) and Dynamic Sparse Training (DST).
- **High Significance:** The focus on data-constrained scaling is highly relevant to current industry trends as high-quality human data becomes increasingly scarce.
- **Valuable Observations:** The finding that sparsity delays data saturation ($R^*_d$) is an interesting and potentially impactful insight for multi-epoch training.

----

**Weaknesses**

- **Speculative Efficiency Metrics:** The primary reliance on theoretical FLOPs is problematic. In modern hardware, DST savings often fail to translate into wall-clock speedups due to indexing overhead and memory-bound operations.
- **Lack of Runtime Analysis:** The absence of actual latency or wall-clock data (e.g., A100/H100 hours) makes the "efficiency" claims feel theoretical rather than practical.
- **Limited Practical Justification:** The paper focuses on DST but does not sufficiently justify why researchers should prefer it over Mixture-of-Experts (MoE), which is currently the industry standard for hardware compatibility.
- **Incremental Originality:** The proposed framework is primarily an incremental integration of established scaling laws (Hoffmann et al., 2022; Muennighoff et al. 2023; Frantar et al. 2024) - augmented by new formulations for sparsity-dependent data and model saturation thresholds. While these additions provide useful granularity, the core conceptual components remain largely derived from prior work

---

> ### Author Rebuttal · Authors · 2026-03-30
>
> Thank you for the  review. We address your concerns under three perspectives:
>
> ### 1. Efficiency
>
> > W1. Speculative Efficiency Metrics..
>
> > W2. Lack of Runtime..
>
> > Q1. Wall-clock...
>
> Theoretical FLOPs do not directly imply wall-clock speedups for unstructured sparsity on standard GPUs. Still, sparse efficiency is not purely theoretical: recent works show **progress** on CPUs [1], emerging AI accelerators [2], and GPU-friendly sparse patterns such as block sparsity [3]. These works suggest that the gap between theoretical and realized sparse efficiency is narrowing, although efficient unstructured sparse training on standard GPUs remains limited.
>
> We therefore study how sparsity changes data utilization under realistic efficiency uncertainty. Figure 5 considers two extreme cost models: **sparse FLOPs**, which count only non-zero operations, and **dense FLOPs**, which count the cost of the dense-equivalent model. These correspond to optimistic and conservative efficiency assumptions, respectively. Under both, the qualitative conclusion is the same: as the training budget increases, sparse models eventually become preferable, with the transition occurring earlier under the optimistic model.
>
> To make the practical discussion more concrete, we include an **inference runtime analysis**. Although our paper studies unstructured sparsity, GPU acceleration is much more mature for block-sparse execution, so we use BLaST [3] under a 16×16 block-sparse setting as a proxy. For our 1B-scale model, the 16×16 block-sparse model achieves validation loss close to the corresponding unstructured sparse model and also provides practical speedups (Table 1). While this does not replace end-to-end wall-clock measurements for unstructured sparse training, it provides a useful practical reference.
>
> We believe it is meaningful to study DST in the data-constrained regime even before practical sparse training speedups are fully realized.
>
> Table 1. Runtime Analysis.
> |Sparsity(%)|Latency(ms)|Throughput (TFLOPS)|
> |-|-|-|
> |0|256.94|236.41|
> |50|219.72|276.45|
> |70|182.07|333.63|
> |80|160.00|379.65|
> |90|133.57|454.77|
> |95|120.00|506.18|
>
> ### 2. Why DST instead of MoE?
>
> > W3. Limited Practical Justification: ..
>
> > Q2. MoE Comparison: ..
>
> We agree that MoE is currently the dominant sparse architecture for large-scale deployment, and we do not claim that practitioners should replace it with DST. Our point is narrower: DST is the more appropriate setting for the question studied here. We believe this for three reasons:
>
> **(1) Cleaner setting for the scientific question.**
> This paper studies how sparse training interacts with repeated data in the data-constrained regime. DST is better suited to this question because it isolates sparse optimization more directly, without MoE-specific confounds such as routing, expert specialization, load balancing, and token dispatch.
>
> **(2) Potential practical relevance.**
> DST and MoE should not be viewed as competing alternatives. Rather, DST is worth studying as a direction complementary to MoE. In settings where MoE relies on many experts, DST may in some cases have a lighter weight and optimizer-state footprint. We do not present this as an argument against MoE, but as one reason DST remains practically meaningful to study.
>
> **(3) Relevance beyond DST itself.**
> While we do not claim that our conclusions transfer directly to MoE, the framework developed here may still be informative for broader sparse-training settings, including future studies of sparse optimization within MoE.
>
> ### 3. Novelty
>
> > W4. Incremental Originality: …
>
> We respectfully disagree that the originality is merely incremental.  Our novelty lies not in treating sparsity as just another scaling variable, but in identifying a data-constrained regime where sparsity and repeated data reuse must be modeled jointly.
> More concretely, our contribution is novel in three ways:
>
>  (1) **A motivated setting**: we study sparse optimization under repeated data reuse because sparse and dense training may differ in how they utilize repeated data.
>
> (2) **A new interaction**: sparsity changes repeated-data utilization itself, so sparsity and data reuse cannot be treated as independent variables.
>
> (3) **A harder experimental and modeling problem**: our setting jointly involves model size, unique token count, sparsity, and repetition, substantially expanding both the experimental space and the complexity of the analysis.
>
> ### 4. Minor…
>
> We thank the reviewer for pointing this out and will revise accordingly.
>
> [1] A comprehensive performance study of large language models on novel ai accelerators. arXiv'23.
>
> [2] SparseProp: Efficient Sparse Backpropagation for Faster Training of Neural Networks, ICML'23.
>
> [3] BLaST: High Performance Inference and Pretraining using BLock Sparse Transformers. arXiv'25.

---

> > ### Author Rebuttal · Reviewer_QVg7 · 2026-04-02
> >
> > Thank you to the authors for answering my questions. I'm satisfied with the rebuttal and will increase my score to 5.

---

> > > ### Author Response · Authors · 2026-04-05
> > >
> > > Dear Reviewer QVg7,
> > >
> > > Thank you very much for taking the time to read our paper and rebuttal. We truly appreciate your constructive comments throughout the review process, which helped improve the overall quality of our work. We are grateful that our responses were able to address your questions.
> > >
> > > Best wishes,
> > >
> > > authors

---

### Official Review · Reviewer_41EN · 2026-03-18

**Soundness:** 3
**Presentation:** 3
**Significance:** 2
**Originality:** 3
**Overall Recommendation:** 4
**Confidence:** 3

**Summary:**

This paper studies the scaling behavior of sparse models, trained via dynamic sparse training, under data-limited conditions. It reports two key empirical findings: (1) given a fixed budget of non-zero parameters, increasing sparsity initially improves validation loss but eventually reaches a point of diminishing returns, and (2) higher sparsity postpones the onset of performance degradation due to repeated exposure to the same data. Building on these observations, the authors argue that sparsity alters the optimal allocation of compute by favoring longer training with data reuse, leading to a reduced parameter-to-token ratio and enhanced inference efficiency.

**Compliance With Llm Reviewing Policy:**

Affirmed.

**Key Questions For Authors:**

Q1. In Figure 1, increasing sparsity appears to reduce validation loss up to a certain point (around 50%). Do the authors have any conjecture for why this occurs? In particular, sparse parameters correspond to zero-valued weights that do not affect forward propagation, thus it is not immediately clear why increasing sparsity would improve validation loss.

Q2. The paper argues that increasing sparsity delays data saturation, yet the effect appears U-shaped (Figure 2). Do the authors have a high-level intuition for this behavior? Why does excessive sparsity begin to slightly harm performance?

Q3. In Figure 4, introducing sparsity into a dense network seems to allow more effective data reuse compared to a dense model. Do the authors have any intuition for this result? Sparsity does not change factors such as tokenization or optimization, which typically dominate training dynamics, so it is somewhat surprising that simply introducing sparsity would delay overfitting to the training data, as the sparsity mask does not change across epochs during training.

**Limitations:**

yes

**Strengths And Weaknesses:**

# Strengths
The paper is clearly written and well organized, making the presentation easy to follow. The study of scaling laws for sparse models in a data-constrained regime, and its comparison with established scaling laws for dense models, is an interesting direction. Understanding how sparse training interacts with known scaling behaviors could be valuable for the pre-training community.

# Weaknesses
The paper would benefit from a clearer articulation of the broader motivation. In particular, it would be helpful to explain why investigating the training dynamics of models trained with dynamic sparse training (DST) is important. In the current large-scale training landscape, sparsity is often associated with Mixture-of-Experts (MoE) architectures rather than DST. As a result, it is not immediately clear why it is necessary to investigate scaling laws for DST-based sparse models.
More broadly, the paper primarily reports empirical observations without offering much intuition about the mechanisms behind them. The work would be more impactful if the authors provided hypotheses or theoretical insights explaining the observed behaviors.

---

> ### Author Rebuttal · Authors · 2026-03-30
>
> Thank you for the review! We respond to your comments and questions in detail below.
>
>
> >  W1:…why ... investigate scaling laws for DST-based sparse models...offering much intuition about the mechanisms...
>
> We thank the reviewer for this helpful comment. We agree that our broader motivation should be stated more clearly.
> Our goal in this paper is not to analyze a particular sparse architecture, but to study a more fundamental question: how sparse training interacts with repeated data in the data-constrained regime.
>
> DST is the appropriate object of study here because it **introduces sparsity directly at the training and optimization level**: only a subset of parameters is updated at each step, and this subset evolves during training. This directly affects how repeated data interacts with parameter updates, which is precisely the mechanism of interest in our setting. By contrast, MoE introduces sparsity at the architectural level, together with additional factors such as routing, expert specialization, and load balancing, making it harder to isolate the effect of sparse training itself on repeated-data scaling.
>
> We therefore view DST Llama as a cleaner first setting for studying sparse training under data reuse. We do not claim that our conclusions directly transfer to MoE, but we believe the qualitative insights and framework developed here may still be relevant more broadly. Extending this analysis to study how sparse optimization interacts with MoE under data constraints is a promising direction for future work, but would require substantially larger-scale experiments and is beyond the scope of the current paper.
>
> We agree that providing more intuition about the underlying mechanisms would strengthen the paper, and we thank the reviewer for this helpful suggestion. In the responses below, we provide our current high-level intuition for several key empirical observations, and we will incorporate these explanations more explicitly into the revision.
>
> > Q1: In Figure 1, increasing sparsity appears to reduce validation loss up to a certain point (around 50%). ..
>
> A key point in Figure 1 is that each panel fixes the **number of active (non-zero) parameters**, rather than the total number of parameters. As a result, increasing sparsity in this figure does not mean that fewer active parameters are used in forward propagation. Instead, the same active parameter budget is selected from a larger total parameter pool.
>
> We therefore do not interpret Figure 1 as showing that simply removing parameters improves validation loss. Rather, our current intuition is that, under a fixed active compute budget, moderate sparsity may help because it allows the model to choose its active connections from a larger model, giving greater flexibility. At the same time, if sparsity becomes too high, optimization may become more difficult and effective connectivity may be reduced, which could explain why the benefit eventually diminishes. We will clarify this point more explicitly in the revision.
>
> > Q2: … U-shaped (Figure 2)…a high-level intuition … Why does excessive sparsity begin to slightly harm performance?
> Our current intuition is that the U-shaped behavior reflects a trade-off between delayed saturation and learning efficiency. Under dense training, every step updates essentially the same full set of weights. When the same limited data is revisited many times, this quickly leads to diminishing returns.
>
> At moderate sparsity, only a subset of parameters is active at each step, and this subset changes over training. Repeated passes over the same data are therefore spread across **different subsets of connections over time, rather than being concentrated on one fixed set of weights**. This makes repeated data reuse more effective and delays saturation.
>
> However, when **sparsity becomes too high, learning becomes too fragmented**. The same update budget is spread over a much large sparse parameter pool. This makes it harder for the model to integrate what it learns efficiently. At that point, the benefit of delayed saturation is outweighed by reduced learning efficiency, so performance begins to degrade.
>
> > Q3: In Figure 4, introducing sparsity into a dense network seems to allow more effective data reuse ...
>
> A key point is that, in our DST setting, the **sparsity mask is dynamic** rather than fixed across epochs. We believe this is central to the behavior shown in Figure 4. Under dense training, repeated exposure to limited data keeps refining largely the same set of connections, so the benefit of additional passes diminishes relatively quickly. Under DST, by contrast, the active sparse subnetwork evolves during training, so repeated passes are distributed over changing subsets of connections rather than repeatedly refining one fixed sparse structure. This can be viewed heuristically as a search over different sparse connectivity patterns, which may make repeated data reuse more effective and delay saturation.

---

> > ### Author Rebuttal · Reviewer_41EN · 2026-04-06
> >
> > Thank you for the clear and constructive rebuttal. It clarified the paper’s motivation, especially why DST is the right setting for studying sparse training under repeated data, and it also provided helpful high-level intuitions for the main empirical trends. My concerns are therefore partially resolved.
> >
> > I still have one follow-up question.
> >
> > For the revised paper, could the authors make it especially clear which parts of the discussion are intended as high-level intuition versus claims that are directly established by the current experiments?

---

> > > ### Author Response · Authors · 2026-04-06
> > >
> > > Dear Reviewer 41EN，
> > >
> > > Thank you for this helpful follow-up. We agree that the revised paper should distinguish more clearly between empirical findings and high-level intuition.
> > >
> > > More specifically, in the revised paper, we will separate more clearly what is directly shown by the experiments from how we interpret those results. As concrete examples, we will revise the discussions of Figures 1, 2, and 4 as follows: the empirical findings are lower validation loss at moderate sparsity, the observed U-shaped dependence on sparsity, and improved data reuse under DST relative to dense training, whereas the ideas involving a larger parameter pool, delayed saturation induced by sparse optimization, learning efficiency, and evolving subnetworks/connectivity will be presented as intuition for these patterns.
> > >
> > > Throughout the revision, we will also adjust the wording accordingly, using words such as “the experiments show that ...” for empirical claims, and “one possible explanation is ...” or “we interpret this as ...” when discussing intuition. We believe this will make the discussion more precise and easier to follow.
> > >
> > > Finally, we also sincerely thank the reviewer for the careful reading of both our paper and rebuttal, and for the thoughtful suggestions, which have helped us improve the clarity and overall quality of the revised paper.
> > >
> > > Best wishes,
> > >
> > > authors

---

### Decision · Program_Chairs · 2026-04-30

**Decision:**

Accept (regular)

**Comment:**

This paper studies Dynamic Sparse Training (DST) in data-constrained regimes and proposes a scaling law that captures the interaction between sparsity and data repetition. Reviewers appreciated the importance of the problem, the extensive empirical study, and the clear presentation. In particular, the finding that sparsity delays data saturation and improves data efficiency was viewed as interesting and potentially impactful.

During the rebuttal, the authors addressed most concerns by clarifying the motivation for DST, adding additional experiments, and providing preliminary efficiency analysis. As a result, several reviewers raised their scores.

Still, some limitations remain. This paper lacks a deeper theoretical understanding of why sparsity delays data saturation. Also, the paper can be strengthened by a more extensive evaluation of downstream performance. In addition, based on my own reading, the proposed scaling law does not appear to be highly accurate (e.g., $R^2 < 99%$). This is perhaps expected, as recent work has suggested that the form of data-constrained scaling laws assumed in this paper may not be fully accurate, for example due to the assumption that $R^*$ is constant [1].

However, these do not outweigh the overall contributions of the paper. Therefore, we recommend acceptance.

[1] Larger Datasets Can Be Repeated More: A Theoretical Analysis of Multi-Epoch Scaling in Linear Regression. ICLR 2026.